# Warping the Space: Weight Space Rotation for Class-Incremental Few-Shot Learning

**Do-Yeon Kim**[1], **Dong-Jun Han**[2*], **Jun Seo**[3], **Jaekyun Moon**[1]
[1] Korea Advanced Institute of Science & Technology, [2] Purdue Univirsity, [3] LG AI Research
dy.kim@kaist.ac.kr, han762@purdue.edu,
jun.seo@lgresearch.ai, jmoon@kaist.edu

## Abstract

Class-incremental few-shot learning, where new sets of classes are provided sequentially with only a few training samples, presents a great challenge due to catastrophic forgetting of old knowledge and overfitting caused by lack of data. During finetuning on new classes, the performance on previous classes deteriorates quickly even when only a small fraction of parameters are updated, since the previous knowledge is broadly associated with most of the model parameters in the original parameter space. In this paper, we introduce WaRP, the *weight space rotation process*, which transforms the original parameter space into a new space so that we can push most of the previous knowledge compactly into only a few important parameters. By properly identifying and freezing these key parameters in the new weight space, we can finetune the remaining parameters without affecting the knowledge of previous classes. As a result, WaRP provides an additional room for the model to effectively learn new classes in future incremental sessions. Experimental results confirm the effectiveness of our solution and show the improved performance over the state-of-the-art methods.

## 1 Introduction

Humans can easily acquire new concepts while preserving old experiences continually over the course of their life span. With a growing desire to imitate such ability, incremental or continual learning has been brought into the spotlight in the AI community recently (Hung et al., 2019; Wortsman et al., 2020; Saha et al., 2021). Here, due to storage and privacy constraints, it is impractical to save all the training samples of previous tasks during the training process (Desai et al., 2021). The most challenging issue in this setup is to preserve the knowledge of previous tasks against catastrophic forgetting (Serra et al., 2018). More recently, an increasing need for such learning capability when dealing with rare data (e.g., military image, medical data, photos of rare animals) has encouraged many researchers to focus on a more challenging setup, known as class-incremental few-shot learning (CIFSL). In CIFSL, each task consists of only a few training samples, making the problem much harder as we must additionally handle the severe overfitting issue caused by lack of training data.

Prior works on CIFSL typically take the following two steps: (i) pretraining the model on the first task (base classes) and (ii) adapting the pretrained model to new classes (novel classes) in each training session (e.g., via finetuning), assuming that the first task contains a sufficiently large number of training samples. One recent work, F2M (Shi et al., 2021), tries to find the flat local minima during the pretraining stage and then finetunes the model within this flat area for learning novel classes so that the forgetting issue could be resolved. Another line of work, named FSLL (Mazumder et al., 2021), mitigates the forgetting issue by keeping some important parameters frozen and finetunes only the remaining trainable parameters during the incremental learning sessions. Several other works adopt different strategies to learn new classes well during incremental sessions without forgetting (Tao et al., 2020; Cheraghian et al., 2021a; Zhang et al., 2021; Kukleva et al., 2021; Chen & Lee, 2021; Akyürek et al., 2022).

**Motivation.** One key strategy shared among some of these prior works is: model update is performed in the *original parameter space*, which is defined here with the *standard basis*. To illustrate the

---

*Corresponding author.

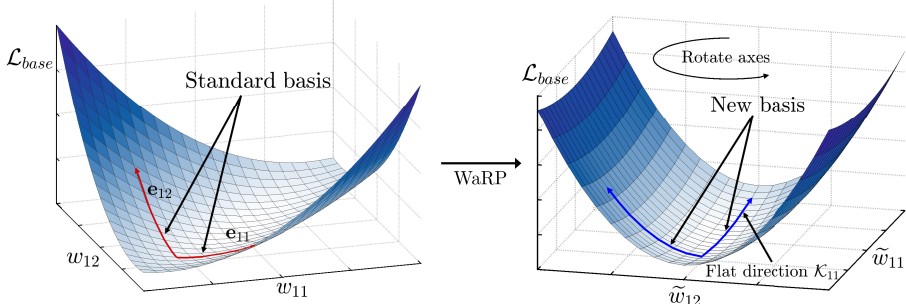

Figure 1: Visualization of loss landscape for base classes after pretraining. **Left:** red dashed arrow lines refer to directions of standard basis. Finetuning along either of these directions at each incremental session can adversely affect the loss. In this original space, it is generally challenging to capture the flat directions directly. **Right:** weight space rotation with desired basis (blue dashed lines) provides us with flat direction $\mathcal{K}_{11}$; following this direction, we can finetune the model without performance loss on the base classes. The performance on the novel classes are also preserved by accumulating and freezing the important parameters obtained at each session.

concept of standard basis in this context, consider an example with $2 \times 2$ weight matrix $W$ for a specific layer. By defining $w_{ij}$ as the $(i, j)$-th parameter of $W$, we can unroll $W$ as follows:

$$W = w_{11} \begin{bmatrix} 1 & 0 \\ 0 & 0 \end{bmatrix} + w_{12} \begin{bmatrix} 0 & 1 \\ 0 & 0 \end{bmatrix} + w_{21} \begin{bmatrix} 0 & 0 \\ 1 & 0 \end{bmatrix} + w_{22} \begin{bmatrix} 0 & 0 \\ 0 & 1 \end{bmatrix}$$

$$= w_{11}\mathbf{e}_{11} + w_{12}\mathbf{e}_{12} + w_{21}\mathbf{e}_{21} + w_{22}\mathbf{e}_{22}. \tag{1}$$

Here, we denote $\mathbf{E} = \{\mathbf{e}_{11}, \mathbf{e}_{12}, \mathbf{e}_{21}, \mathbf{e}_{22}\}$ as standard basis, upon which the weight space is parameterized by $\mathbf{w} = \{w_{11}, w_{12}, w_{21}, w_{22}\}$. Neural networks are typically implemented in the weight space spanned by $\mathbf{E}$, and thus the learning processes (e.g. gradient computation, parameter update) are performed with respect to the parameters in $\mathbf{w}$ as shown in Figure 1. However, our empirical observation in Figure 2, which showcases the effect of simple finetuning in two different spaces, suggests that the accuracies of previous classes are extremely vulnerable to model finetuning in the original space, even when we finetune only a small fraction of parameters; finetuning only 3% of model parameters (freeze 97%) throughout the incremental sessions significantly degrades the performance on base classes. This implies that the previous knowledge is more broadly associated with, to a certain extent, most of the model parameters in the original space. Motivated by this, we pose the following question: *Can we find another basis, i.e., a new weight parameter space, such that we can push most of the previous knowledge compactly into only a few key parameters?*

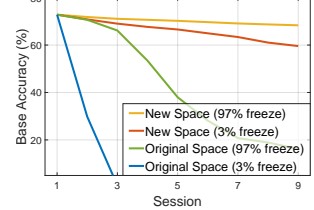

Figure 2: Original vs. New space.

**Contributions.** In this paper, we introduce the concept of **W**eight sp**a**ce **R**otation **P**rocess (WaRP) that provides a solution to the above question. By viewing the weight matrix of a neural network from a different perspective, WaRP transforms the original parameter space into a new space. For any given orthonormal set of matrices, we can reparameterize the weight in (1) as

$$W = \widetilde{w}_{11}\mathcal{K}_{11} + \widetilde{w}_{12}\mathcal{K}_{12} + \widetilde{w}_{21}\mathcal{K}_{21} + \widetilde{w}_{22}\mathcal{K}_{22} \tag{2}$$

where $\mathcal{B} = \{\mathcal{K}_{11}, \mathcal{K}_{12}, \mathcal{K}_{21}, \mathcal{K}_{22}\}$ is a newly constructed basis that consists of orthonormal matrices and $\widetilde{w}_{ij}$ is the weight parameter that is reparameterized according to $\mathcal{K}_{ij}$. As shown in Figure 1, we can always rotate the axes by which the weight is represented for an arbitrary basis $\mathcal{B}$. Here, if $\widetilde{w}_{12}$ in (2) plays a key role in determining the output of the layer, then $\{\mathcal{K}_{11}, \mathcal{K}_{21}, \mathcal{K}_{22}\}$ can be viewed as flat directions. Thus, finetuning can be performed *along the flat directions* by freezing the important parameter $\widetilde{w}_{12}$ in the new space, which effectively preserves the previous knowledge (Figure 2). Figure 1 provides high-level descriptions of WaRP and our finetuning process along the flat direction.

By introducing the concept of WaRP, we propose a new strategy to construct an appropriate basis $\mathcal{B}$ using the low-rankness property of activation, and redefine the model parameters in this new space so that we can push most of the knowledge of base classes compactly into only a few parameters. Once we construct a new weight space, we identify and freeze the important parameters for keeping the knowledge of previous classes at the end of each training session, and finetune only the remaining

parameters in the next session without performance degradation on previous knowledge. With a careful construction of new basis and a proper criterion for identifying important parameters, we can preserve the knowledge of base classes after finetuning the model on novel classes at each incremental session. The knowledge of novel classes can be also preserved by accumulating the important parameters and keeping them frozen. It turns out that WaRP is easy to implement and the proposed score criterion is efficient to compute. Experimental results on various benchmarks show the improved performance of our method over the state-of-the-art baselines.

## 2    WEIGHT SPACE ROTATION

Before describing our solution for CIFSL, we first introduce the general concept of WaRP proposed in this paper. To begin with, we view the weight matrix $W \in \mathbb{R}^{m \times n}$ as a linear combination of orthonormal matrices $\{\mathbf{e}_{ij}\}_{i,j}$, where $\mathbf{e}_{ij} \in \mathbb{R}^{m \times n}$ is defined as a matrix having one for the $(i,j)$-th element and zeros for others. We can rewrite $W$ as

$$W = \sum_{i=1}^{m} \sum_{j=1}^{n} w_{ij} \mathbf{e}_{ij} \tag{3}$$

where $w_{ij}$ denotes the $(i,j)$-th element of $W$ which is the coefficient of $\mathbf{e}_{ij}$. Since neural networks are parameterized by coefficients $\mathbf{w} := \{w_{ij}\}_{i,j}$ in practical deep learning implementations, typical learning operations such as gradient descent are conducted on the weight space spanned by the standard basis $\mathbf{E} := \{\mathbf{e}_{ij}\}_{i,j}$. However, the learned knowledge of previous tasks is likely to be associated with most of the parameters in $\mathbf{w}$ (i.e., most directions in standard basis $\mathbf{E}$); updating a small fraction of selected parameters could give rise to significant performance degradation on previous classes as shown in Figure 2, suggesting that the previous performance is highly vulnerable to finetuning in the original space.

In this paper, we extend the above linear algebraic interpretation of the weight matrix (based on standard basis $\mathbf{E}$) using an arbitrary basis $\mathcal{B} = \{\mathcal{K}_{ij} \in \mathbb{R}^{m \times n} | i \in [m], j \in [n]\}$ that spans $\mathbb{R}^{m \times n}$. Specifically, let $\mathcal{B}$ be an arbitrary set of orthonormal matrices satisfying: (i) $\langle \mathcal{K}, \mathcal{K}' \rangle = 0$ if $\mathcal{K} \neq \mathcal{K}'$ for $\mathcal{K}, \mathcal{K}' \in \mathcal{B}$, and (ii) $\|\mathcal{K}\| = \sqrt{\langle \mathcal{K}, \mathcal{K} \rangle} = 1$ for all $\mathcal{K} \in \mathcal{B}$, where $\| \cdot \|$ and $\langle \cdot, \cdot \rangle$ are the norm and inner product properly defined in $\mathbb{R}^{m \times n}$, respectively. Since a vector space $\mathcal{H} := \mathbb{R}^{m \times n}$ equipped with the inner product operation $\langle A, B \rangle := \text{trace}(A^\top B)$ (for $A, B \in \mathcal{H}$) is Hilbert space, one can always rewrite $W \in \mathcal{H}$ as a linear combination of orthonormal matrices in $\mathcal{B}$ as follows:

$$W = \sum_{\mathcal{K} \in \mathcal{B}} \langle W, \mathcal{K} \rangle \mathcal{K} = \sum_{i=1}^{m} \sum_{j=1}^{n} \langle W, \mathcal{K}_{ij} \rangle \mathcal{K}_{ij}. \tag{4}$$

Note that this equation generalizes (3) to the case with an arbitrary basis $\mathcal{B}$; the special case with $\mathcal{K}_{ij} = \mathbf{e}_{ij}$ (i.e., $\mathcal{B} = \mathbf{E}$) reduces to the result in equation (3). Thus for any orthonormal set of matrices $\mathcal{B}$ spanning $\mathbb{R}^{m \times n}$, we can always reparameterize the layer of the model with $\widetilde{\mathbf{w}} := \{\langle W, \mathcal{K} \rangle\}_{\mathcal{K} \in \mathcal{B}}$ in place of $\mathbf{w}$. In the next section, we describe how WaRP can address forgetting in CIFSL.

## 3    PROPOSED ALGORITHM FOR CIFSL

**Problem Setup and Notations.** The main goal of CIFSL is to incrementally train the model on a new set of classes at each training session, while avoiding forgetting of old classes. Let $\mathcal{T}_k$ be the task given in $k$-th session with tuples of training sample and its class label, i.e., $\mathcal{T}_k = \{(x_t^k, y_t^k)\}_{t=1}^{N_k}$, where $N_k$ denotes the number of samples in $\mathcal{T}_k$. For simplicity, we omit super/subscripts of $x_t^k$ and $y_t^k$ if these can be clearly referred from the context. By defining $n_s$ as the total number of training sessions, we assume that the corresponding sets of categories in each task, denoted by $\mathcal{C}_1, \ldots, \mathcal{C}_{n_s}$, are completely disjoint. As in previous works (Tao et al., 2020; Shi et al., 2021), we consider a setup where a sufficiently large amount training data are available in the first task $\mathcal{T}_1$. In contrast, the subsequent tasks $\mathcal{T}_2 \ldots, \mathcal{T}_{n_s}$ consist of only a small number of samples per class, e.g., 5 samples per class. We consider a practical setup where all training samples of previous tasks cannot be accessed during training due to privacy and storage constraints. We use the term 'task' and 'session' interchangeably throughout the paper. We let $W_l \in \mathbb{R}^{m_l \times n_l}$ be the weight matrix of $l$-th layer in the neural network. We discard the layer index $l$ and use $W \in \mathbb{R}^{m \times n}$ for weight whenever the context obviates it. $I_n$ denotes the $n \times n$ identity matrix, $[n]$ denotes the set $\{1, \ldots n\}$, and $\{a_i\}_i$ or $\{a_i\}_{i \in \mathcal{I}}$ represents $\{a_i | i \in \mathcal{I}\}$ for some index set $\mathcal{I}$.

**Overview of Approach.** Given the model trained on the first task $\mathcal{T}_1$, our first step is to construct a new basis via WaRP on which most directions are flat. Once new basis is constructed in the first session, this new basis is fixed throughout the whole remaining (incremental) sessions. At the end of each session, we identify the important parameters based on the scoring criterion and freeze them. In the next session, we finetune only the remaining parameters, namely, the directions of flat axes, and the same process is repeated for the future tasks. Throughout the learning process, the important parameters are accumulated and kept frozen to preserve the knowledge of both novel and base classes.

In the following, we start by describing how to construct an appropriate basis via WaRP for CIFSL and how to implement it in practice (Sec. 3.1). Then we describe our criterion for identifying the important parameters in each incremental session (Sec. 3.2). Finally, we describe the overall procedure of our algorithm for CIFSL with details (Sec. 3.3).

## 3.1 Constructing New Basis for Weight Space Rotation

To find an appropriate basis to reparameterize $W \in \mathbb{R}^{m \times n}$, we start by expressing the basis by using a pair of unitary matrices. Given any two unitary matrices $V \in \mathbb{R}^{m \times m}$, $U \in \mathbb{R}^{n \times n}$, it can be seen that the set $\mathcal{B} = \{\mathcal{K}_{ij} := \mathbf{v}_i \mathbf{u}_j^\top \in \mathbb{R}^{m \times n} | i \in [m], j \in [n]\}$ is orthonormal, where $\mathbf{v}_i$ and $\mathbf{u}_j$ denote the $i$-th and $j$-th column vectors of $V$ and $U$, respectively. Thus from (4), we can rewrite $W$ as

$$W = \sum_{i=1}^{m} \sum_{j=1}^{n} \langle W, \mathcal{K}_{ij} \rangle \mathcal{K}_{ij} = \sum_{i=1}^{m} \sum_{j=1}^{n} \langle W, \mathbf{v}_i \mathbf{u}_j^\top \rangle \mathbf{v}_i \mathbf{u}_j^\top = \sum_{i=1}^{m} \sum_{j=1}^{n} \widetilde{w}_{ij} \mathbf{v}_i \mathbf{u}_j^\top, \tag{5}$$

where $\widetilde{w}_{ij} := \langle W, \mathbf{v}_i \mathbf{u}_j^\top \rangle$. In other words, $W$ is reparameterized by $\{\widetilde{w}_{ij}\}_{i,j}$ using the basis $\{\mathbf{v}_i \mathbf{u}_j^\top\}_{i,j}$ constructed with arbitrary two unitary matrices $V$ and $U$. Now the question is, how to design appropriate $V$ and $U$ to make most of the elements in the basis to be flat directions?

For a specific layer $l$, we define a matrix $\Phi_l = [\phi_l(x_1), \phi_l(x_2), \ldots, \phi_l(x_{N_1})]$ where $\phi_l(x_t)$ denotes the activation of the $l$-th layer for the $t$-th sample in the first task $\mathcal{T}_1$. Then the singular value decomposition of covariance of activations is given by

$$\Phi_l \Phi_l^\top = U_l \Sigma_l U_l^\top, \tag{6}$$

where $\Sigma_l$ is a diagonal matrix having singular values on its diagonal and $U_l$ is the corresponding unitary matrix. Here, note that the activations of deep neural networks are known to have low-rankness property as shown in (Suzuki et al., 2020b; Wang et al., 2021), i.e., singular values decrease rapidly and most of the singular values are close to 0. Thanks to this low-rankness property, by adopting $U = U_l$ in (5) for the $l$-th layer $W_l$ and defining $\widetilde{w}_{ij}^l = \langle W^l, \mathbf{v}_i \mathbf{u}_j^\top \rangle$, we have

$$W_l \phi_l(x) = \sum_{i=1}^{m} \sum_{j=1}^{n} \widetilde{w}_{ij}^l \underbrace{\mathbf{v}_i \mathbf{u}_j^\top \phi_l(x)}_{\approx \mathbf{0} \text{ for most } (i,j) \text{ pairs by low-rankness.}} \tag{7}$$

for any unitary matrix $V$ and any sample $x$ in $\mathcal{T}_1$. Namely, the effect of activation $\phi_l(x)$ is highly likely to be negligible in most directions in $\mathcal{B} = \{\mathbf{v}_i \mathbf{u}_j^\top\}_{i,j}$ during forward propagation. In other words, most directions in $\mathcal{B}$ are flat and thus only a few directions and their corresponding parameters are meaningful. So if we can properly identify the important parameters among $\{\widetilde{w}_{ij}\}_{i,j}$ and freeze them, finetuning along the remaining directions (for which $\mathbf{v}_i \mathbf{u}_j^\top \phi_l(x) \approx \mathbf{0}$) does rarely change the output $W_l \phi_l(x)$ for $x$ in $\mathcal{T}_1$. This implies that the feature embedding of previous classes can be preserved by freezing only a small fraction of parameters in this new weight space.

**Implementation.** Note that the actual implementation of WaRP in (5) turns out to be simple in practice. By using the fact that $VV^\top = I_m$ and $UU^\top = I_n$, we have $W = VV^\top WUU^\top = V\widetilde{W}U^\top$ where $\widetilde{W} := V^\top WU$. From the definition above, $\widetilde{w}_{ij}$ is $(i,j)$-th element of $\widetilde{W}$. Thus to apply WaRP, we just multiply unitary matrices to $W$ and reparameterize the layer with $\widetilde{W}$ while $V$, $U$ remain fixed. Throughout the algorithm, we set $V = I_m$ which we found to be a valid choice.

**Convolutional Case.** Implementing WaRP for convolutional layers is almost same with the case of fully-connected layers except for flattening the feature maps and reshaping the weight/unitary matrices, to make the pipelines workable as in fully-connected layer. The detailed implementation for convolutional layer can be found in Appendix A.2.

## 3.2 Identifying Important Parameters

In the new weight space constructed in Sec. 3.1, our scheme identifies the important parameters at the end of each training session and keep freezing them throughout the training process. The remaining trainable parameters (except the accumulated important parameters) are finetuned in the next session. In this subsection, we describe our strategy for selecting the important parameters in each session.

To identify the trainable parameters (i.e., flat directions), at the end of each $k$-th session, we take the score criterion that is well compatible with our new space constructed by WaRP. By the chain rule, taking the derivative of the loss function, denoted by $L$, with respect to the parameter $\widetilde{w}_{ij}^{l}$ yields:

$$dL/d\widetilde{w}_{ij}^{l} = dz_l^{\top} \underbrace{\mathbf{v}_i \mathbf{u}_j^{\top} \phi_l(x)}_{(a)}, \tag{8}$$

where $dz_l$ denotes the gradient of the loss with respect to the output of the layer $z_l = W_l \phi_l(x)$ for any $x$ in $\mathcal{T}_k$. Interestingly, it turns out that the term $(a)$ in equation (8) also appears in (7). In addition, as shown in (8), the trend of the magnitude of $dL/d\widetilde{w}_{ij}^{l}$ follows the tendency of $(a)$. From these observations, it can be seen that the gradient can indirectly capture the influence of finetuning within each direction $\mathbf{v}_i \mathbf{u}_j^{\top}$ on $\mathcal{T}_k$. Motivated by this, we consider the following score criterion:

$$s_{ij}^{l} := \text{importance score of } \widetilde{w}_{ij}^{l} = \sum_{b \in \mathcal{D}_k^{n_b}} \left| dL_k(b)/d\widetilde{w}_{ij}^{l} \right| \tag{9}$$

where $L_k(b)$ is the loss computed with batch $b$ in $\mathcal{T}_k$, and $\mathcal{D}_k^{n_b}$ is a set that consists of $N_k/n_b$ batches with size $n_b$ in $\mathcal{T}_k$. In words, the score is computed by summing up the magnitudes of the gradients using the samples in $\mathcal{T}_k$. Here, if $s_{ij}^{l}$ is small, then $\Delta\widetilde{w}_{ij}\mathbf{v}_i \mathbf{u}_j^{\top} \phi_l(x)$ is likely to be small where $\Delta\widetilde{w}_{ij}$ simulates the gradient with respect to $\widetilde{w}_{ij}$. Thus, finetuning with respect to this $\widetilde{w}_{ij}$ (following the direction of $\mathbf{v}_i \mathbf{u}_j^{\top}$) on new classes may not distort the feature embedding of *all previous classes*.

**Remarks.** We wish to highlight two points here. First, WaRP in Sec. 3.1 and the score criterion in Sec. 3.2 work in a highly complementary manner; if we use the standard basis, the flat directions are limited and thus the performance could be restricted regardless of the scoring criterion. If we use a different score criterion (e.g. weight magnitude as done in (Mazumder et al., 2021)) other than (9) in the new weight space, we cannot fully exploit the low-rankness as in (8) and thus cannot properly identify flat directions. These results are confirmed via experiments in Sec. 5.3. Secondly, due to the simplicity of implementing WaRP, we can rely on auto-grad computation of gradient with respect to the weight matrix; we can take the derivative with respect to $\widetilde{W}$ rather than with respect to $\widetilde{w}_{ij}$ one by one, to obtain the scores at once. Thus it is efficient to compute.

## 3.3 Overall Procedure for CIFSL

Based on WaRP and the proposed score criterion, we describe the overall procedure of our algorithm.

**First Session ($k = 1$).** We pretrain the embedding network and the linear classifier $W_{\text{cls}}$ by minimizing the cross-entropy loss $L_{\text{CE}}(\mathcal{T}_k) = -\frac{1}{N_k} \sum_{t=1}^{N_k} \sum_{c \in \mathcal{C}_k} \mathbb{1}_{\{y_t^k = c\}} \log\left(\frac{\exp(\gamma_{c,t})}{\sum_{c' \in \cup_{i=1}^{k} \mathcal{C}_i} \exp(\gamma_{c',t})}\right)$ with $k = 1$, where $\mathbb{1}_{\{\cdot\}}$ is an indicator function and $\gamma_{c,t}$ denotes the output of classifier corresponding to class $c$. When pretraining is done, we replace the weight vector of linear classifier $W_{\text{cls}}$ corresponding to class $c$ with the prototype (average of feature embeddings corresponding to class $c$), which we found is a good option for balancing the model performance on base classes and future novel classes.

Now we construct a new basis for each layer according to Sec. 3.1 and apply WaRP. Here, to alleviate the overfitting issue on novel classes during the remaining incremental sessions, we restrict the learnable space by applying WaRP only to the last few layers while keeping other layers fixed throughout the whole training process. Note that once we construct a new basis in the first session, this basis is fixed throughout the remaining sessions and thus model updates are performed in the new/fixed space. See Appendix A.4 for discussions on validity of the new space for novel classes.

At the end of the first session, we identify the important/trainable parameters based on the proposed importance score (9) in Sec. 3.2. We select the parameters with the scores up to top $\alpha \times 100\%$ as important parameters and let the remaining ones as trainable parameters, where $\alpha$ is the parameter

keeping ratio, a hyperparameter defined in range $[0, 1]$. To indicate whether each element of $\widetilde{W}_l$ is important or not, we introduce a 0-1 mask $\mathbf{M}_l \in \mathbb{R}^{m_l \times n_l}$ as follows:

$$(i, j)\text{-th element of } \mathbf{M}_l = \begin{cases} 1 & s_{ij}^l \geq s_\alpha \\ 0 & \text{otherwise} \end{cases}, \forall (i, j) \in [m_l] \times [n_l], \forall \text{ selected layers } l \quad (10)$$

where $s_\alpha$ is the score value corresponding to top $\alpha \times 100\%$ of the scores of all parameters in all selected layers, i.e., in $\{s_{ij}^l | i \in [m_l], j \in [n_l], \text{ selected layers } l\}$. If the $(i, j)$-th element of $\mathbf{M}_l$ is 1, the $(i, j)$-th element of $\widetilde{W}_l$ is regarded as an important parameter.

**Incremental Sessions** ($k \geq 2$). In the beginning of each incremental session $k \geq 2$, we compute the prototypes corresponding to new classes in $\mathcal{C}_k$ and utilize them as a classifier along with the prototypes of the base classes. To preserve the knowledge of previous classes during finetuning, we only update the unimportant parameters using the importance mask $\mathbf{M}_l$ obtained from the previous session. We block the gradient computation with respect to important parameters at each layer $l$ as

$$\widetilde{W}_l \leftarrow stop\_grad\left(\widetilde{W}_l \odot \mathbf{M}_l\right) + \widetilde{W}_l \odot (1 - \mathbf{M}_l) \quad (11)$$

and let $W_l \leftarrow V\widetilde{W}_l U^\top$, where $stop\_grad(\cdot)$ is an operation that detaches the input from the computational graph during loss computation, and $\odot$ denotes element-wise multiplication. Thus, gradient descent with respect to $\widetilde{W}_l$ in (11) leads to model update of the unimportant parameters only. In each incremental session $k$, we finetune the model $\widetilde{W}_l$ using the cross-entropy loss $L_{\text{CE}}(\mathcal{T}_k)$ for each selected layer $l$, while keeping the weights of the classifier fixed during finetuning.

After finetuning is finished on task $\mathcal{T}_k$, we also identify the important parameters for $\mathcal{T}_k$ to be considered for the next session; we obtain the importance mask $M_{l,k}$ using criterion in (10) for a given $\alpha$. To preserve not only the knowledge of $\mathcal{T}_k$ but also of classes in previous sessions, we accumulate important parameters identified so far:

$$M_l \leftarrow 1 - (1 - M_l) \odot (1 - M_{l,k}). \quad (12)$$

All the knowledge encountered so far can be properly preserved by this accumulated mask.

Finally, when each $k$-th session is finished, the prediction is made on the test samples from all encountered classes, i.e., for a given input, the prediction $\hat{y}_{\text{test}}$ is given by $\hat{y}_{\text{test}} = \arg\max_{c' \in \cup_{i=1}^k \mathcal{C}_i} \gamma_{c',t}$. The whole process described above is repeated until the last incremental session, i.e., for $k = 2, 3, \ldots n_s$.

## 4 RELATED WORKS

**Class-Incremental Few-shot Learning.** There have been many attempts to solve class-incremental few-shot learning where only a few labeled samples are available in each task (Cheraghian et al., 2021b; Zhu et al., 2021; Zhou et al., 2022b; Cheraghian et al., 2021a). To address both catastrophic forgetting and overfitting issues in CIFSL, some prior works take meta-learning approaches (Gidaris & Komodakis, 2018; Ren et al., 2019; Yoon et al., 2020). They typically mimic the inference stage during training so that the model can rapidly adapt to new classes. However, these works assume that only one set of classes is given during the incremental session and do not consider a scenario in which "more than one" sets of new classes are consecutively provided. TOPIC (Tao et al., 2020) is the first work that considered more than one incoming sets of classes, providing a cornerstone for future emerging studies (Akyürek et al., 2022; Chen & Lee, 2021; Zhang et al., 2021; Mazumder et al., 2021; Shi et al., 2021; Zhou et al., 2022a). The authors of TOPIC adopted neural gas network to maintain the topology of feature space, preventing forgetting issue. FSLL (Mazumder et al., 2021) addressed the overfitting issue by choosing a few trainable parameters to be updated for finetuning and prevented forgetting issue by imposing additional regularization to minimize the deviation from the pretrained model parameters. However, this additional regularization may limit the learnability of the model on new classes. Unlike many existing works, (Shi et al., 2021; Zhou et al., 2022a) turned their focus on preparing a desired pretrained model in advance suitable for future incremental sessions. Specifically, the authors of F2M (Shi et al., 2021) inject noise to model parameters during pretraining to find the flat local minima, and finetune the model within this flat area during incremental sessions. However, pretraining with noise injection can lead to a longer training time for the model to converge and induce a little compromise on the performance of the pretrained model in return for the flatness of local minima. Compared to existing works where the proposed algorithms are performed in the 'original parameter space' (e.g. freezing/updating the parameters in standard basis as done in FSLL),

Table 1: Accuracy on *mini*ImageNet dataset under 5-way 5-shot incremental few-shot learning setup.

| Method | Session | | | | | | | | |
|---|---|---|---|---|---|---|---|---|---|
| | 1 | 2 | 3 | 4 | 5 | 6 | 7 | 8 | 9 |
| Finetuning | **72.99** | 67.79 | 63.71 | 60.17 | 56.82 | 53.18 | 48.36 | 43.43 | 39.60 |
| iCaRL (Rebuffi et al., 2017) | 71.77 | 61.85 | 58.12 | 54.60 | 51.49 | 48.47 | 45.90 | 44.19 | 42.71 |
| Rebalance (Hou et al., 2019) | 72.30 | 66.37 | 61.00 | 56.93 | 53.31 | 49.93 | 46.47 | 44.13 | 42.19 |
| GPM (Saha et al., 2021) | **72.99** | 68.04 | 64.18 | 60.96 | 58.32 | 55.66 | 52.90 | 51.10 | 49.96 |
| EEIL (Castro et al., 2018) | 61.31 | 46.58 | 44.00 | 37.29 | 33.14 | 27.12 | 24.10 | 21.57 | 19.58 |
| TOPIC (Tao et al., 2020) | 61.31 | 50.09 | 45.17 | 41.16 | 37.48 | 35.52 | 32.19 | 29.46 | 24.42 |
| FSLL (Mazumder et al., 2021) | 66.48 | 61.75 | 58.16 | 54.16 | 51.10 | 48.53 | 46.54 | 44.20 | 42.28 |
| FSLL+SS (Mazumder et al., 2021) | 68.85 | 63.14 | 59.24 | 55.23 | 52.24 | 49.65 | 47.74 | 45.23 | 43.92 |
| CEC (Zhang et al., 2021) | 72.00 | 66.83 | 62.97 | 59.43 | 56.70 | 53.73 | 51.19 | 49.24 | 47.63 |
| F2M (Shi et al., 2021) | 72.05 | 67.47 | 63.16 | 59.70 | 56.71 | 53.77 | 51.11 | 49.21 | 47.84 |
| WaRP (Ours) | **72.99** | **68.10** | **64.31** | **61.30** | **58.64** | **56.08** | **53.40** | **51.72** | **50.65** |

our method explores the 'new parameter space' where the axes are aligned with many flat directions along which the model is stably updated, by reparameterzing the weight as in equation (4). With a careful construction of the new weight space, we can finetune along the *flat directions* by properly identifying and freezing the important parameters at each session, enjoying both advantages of F2M (Shi et al., 2021) and FSLL (Mazumder et al., 2021), without any dedicated process to find the flat region in advance or additional regularization on model parameters. Although not directly targeting CIFSL, there is another work (Liu et al., 2018) that reparameterizes the weight using a rotation matrix for tackling continual learning. Its focus is mainly on a good diagonalization of the Fisher information matrix (FIM) that shows up in the objective function of (Kirkpatrick et al., 2017). In our work, however, the role of rotation is different in that we rotate the axes of loss landscape for identifying/utilizing the flat directions aligned with new basis by further exploiting the low-rankness property.

**Utilizing Low-Rankness Property of Activation.** There are a few works that exploit the low-rankness property of activation in deep neural network (Suzuki et al., 2020b;a; Saha et al., 2021; Wang et al., 2021). This suggests that the inputs are highly concentrated on the subspace spanned by only a few number of certain orthogonal vectors. (Suzuki et al., 2020b; Wang et al., 2021) confirm this property by showing the empirical distribution of singular values at some layers. The authors of (Suzuki et al., 2020a) proposed a network compression scheme by exploiting this property which brings on small number of degrees of freedom. In (Suzuki et al., 2020b), the authors derived the tight generalization bound of deep neural network with the help of high compressibility of the model induced by this property. Another line of works (Saha et al., 2021; Wang et al., 2021) try to solve continual learning problem by utilizing the low-rankness property. At each learning session, they update the model parameters using the gradient orthogonal to the subspace which the previous tasks are concentrated on. Note that our work turns out to generalize the update process of (Saha et al., 2021; Wang et al., 2021) in that if we freeze the parameters of the weight matrix in a column-wise manner, the gradient update of our method can be reduced to that of these works as special cases.

## 5 EXPERIMENTS

### 5.1 EXPERIMENTAL SETUP & DETAILS

**Datasets.** We evaluate our method on three benchmark datasets, CIFAR100 (Krizhevsky et al., 2019), *mini*ImageNet (Vinyals et al., 2016) and CUB200 (Wah et al., 2011), in the CIFSL setting. For both CIFAR100 and *mini*ImageNet, we split the total of 100 classes into 60 classes (for base classes) and 40 classes (for novel classes). The 40 novel classes are divided into 8 different sets having 5 categories each, and allocated to each incremental session following the 5-way 5-shot setting. For CUB200, we split the total of 200 classes into 100 for the base classes and the remaining 100 for the novel classes, where the 100 novel classes are divided into 10 different sets having 10 categories each, following the 10-way 5-shot setting in each incremental session. We follow the same split configuration proposed by TOPIC (Tao et al., 2020) in all datasets.

**Model Architecture.** For the embedding network, following the standard setup in (Tao et al., 2020; Mazumder et al., 2021; Shi et al., 2021), we use ResNet20 for CIFAR100 and use ResNet18 (He et al., 2016) for the others. Additionally, to see how a wider model affects the ability of our method for making additional room for learning new classes, we also conduct experiments with ResNet18[1]

---

[1]The ResNet18 model architecture for CIFAR100 dataset is adopted from `https://github.com/kuangliu/pytorch-cifar/blob/master/models/resnet.py`

Table 2: Accuracy on CUB200 dataset under 10-way 5-shot incremental few-shot learning setup.

| Method | Session | | | | | | | | | | |
|---|---|---|---|---|---|---|---|---|---|---|---|
| | 1 | 2 | 3 | 4 | 5 | 6 | 7 | 8 | 9 | 10 | 11 |
| Finetuning | **77.74** | 73.49 | 68.29 | 60.60 | 51.62 | 34.37 | 25.84 | 23.36 | 19.01 | 13.05 | 12.24 |
| iCaRL (Rebuffi et al., 2017) | 75.95 | 60.90 | 57.65 | 54.51 | 50.83 | 48.21 | 46.95 | 45.74 | 43.21 | 43.01 | 41.27 |
| Rebalance (Hou et al., 2019) | 77.44 | 58.10 | 50.15 | 44.80 | 39.12 | 34.44 | 31.73 | 29.75 | 27.56 | 26.93 | 25.30 |
| GPM (Saha et al., 2021) | **77.74** | 73.92 | 70.49 | 66.38 | 64.51 | 62.03 | 60.76 | 59.10 | 56.55 | 56.02 | 54.82 |
| EEIL (Castro et al., 2018) | 68.68 | 53.63 | 47.91 | 44.20 | 36.30 | 27.46 | 25.93 | 24.70 | 23.95 | 24.13 | 22.11 |
| TOPIC (Tao et al., 2020) | 68.68 | 62.49 | 54.81 | 49.99 | 45.25 | 41.40 | 38.35 | 35.36 | 32.22 | 28.31 | 26.28 |
| FSLL (Mazumder et al., 2021) | 72.77 | 69.33 | 65.51 | 62.66 | 61.10 | 58.65 | 57.78 | 57.26 | 55.59 | 55.39 | 54.21 |
| FSLL+SS (Mazumder et al., 2021) | 75.63 | 71.81 | 68.16 | 64.32 | 62.61 | 60.10 | 58.82 | 58.70 | 56.45 | 56.41 | 55.82 |
| CEC (Zhang et al., 2021) | 75.85 | 71.94 | 68.50 | 63.50 | 62.43 | 58.27 | 57.73 | 55.81 | 54.83 | 53.52 | 52.28 |
| F2M (Shi et al., 2021) | 77.13 | 73.92 | 70.27 | 66.37 | 64.34 | 61.69 | 60.52 | 59.38 | 57.15 | 56.94 | 55.89 |
| WaRP (Ours) | **77.74** | **74.15** | **70.82** | **66.90** | **65.01** | **62.64** | **61.40** | **59.86** | **57.95** | **57.77** | **57.01** |

Table 3: Accuracy on CIFAR100 dataset under 5-way 5-shot incremental few-shot learning setup.

| Method | Session | | | | | | | | |
|---|---|---|---|---|---|---|---|---|---|
| | 1 | 2 | 3 | 4 | 5 | 6 | 7 | 8 | 9 |
| Finetuning | 80.31 | 75.43 | 70.82 | 66.26 | 62.21 | 59.08 | 56.43 | 53.58 | 51.05 |
| GPM (Saha et al., 2021) | 80.31 | 75.54 | 71.27 | 66.97 | 63.63 | 60.48 | 57.92 | 55.57 | 53.18 |
| FSLL (Mazumder et al., 2021) | 80.24 | 74.32 | 69.03 | 64.75 | 61.37 | 58.56 | 56.48 | 54.47 | 52.27 |
| FSLL+SS (Mazumder et al., 2021) | **81.42** | 74.79 | 69.33 | 64.97 | 61.60 | 58.92 | 57.06 | 55.28 | 53.12 |
| CEC (Zhang et al., 2021) | 79.69 | 74.77 | 70.24 | 66.13 | 62.93 | 59.94 | 57.67 | 55.63 | 53.44 |
| F2M (Shi et al., 2021) | 79.78 | 74.93 | 70.68 | 66.58 | 63.45 | 60.51 | 58.30 | 56.20 | 53.99 |
| WaRP (Ours) | 80.31 | **75.86** | **71.87** | **67.58** | **64.39** | **61.34** | **59.15** | **57.10** | **54.74** |

on CIFAR100, which is wider than ResNet20. We randomly initialize the network before training on base classes for both CIFAR100 and *mini*ImageNet. For CUB200, we deploy the Pytorch's built-in pretrained model for initialization as existing works have done.

**Evaluation & Design Parameter.** At each session, we evaluate our method by measuring the accuracy on test samples from all encountered classes so far. We conduct 5 simulations under different random seeds and report average values. Regarding the hyperparameters, the parameter keeping ratio is set to be $\alpha = 0.1$ for all experiments. We apply WaRP to the last 1 or 2 resnet blocks while the remaining parameters including batchnorm layers are fixed after the pretraining stage in the first session. For computing the output of the classifier, we use the standard dot product in the beginning, and replace it with cosine similarity (Gidaris & Komodakis, 2018) after adopting the prototypes as a classifier. More detailed hyperparameter settings can be found in Appendix A.1.

**Baselines.** We consider the following recent methods as baselines: iCaRL (Rebuffi et al., 2017), Rebalance (Hou et al., 2019), GPM (Saha et al., 2021), EEIL (Castro et al., 2018), TOPIC (Tao et al., 2020), FSLL (Mazumder et al., 2021), CEC (Zhang et al., 2021) and F2M (Shi et al., 2021). The performance of these baselines are mostly taken from (Zhang et al., 2021; Shi et al., 2021) for fair comparisons. We also consider simple finetuning, which is equivalent to our method with $\alpha = 0$ using a smaller learning rate.

## 5.2 MAIN EXPERIMENTAL RESULTS

We first report the evaluated accuracy of each scheme on *mini*ImageNet, CUB200 and CIFAR100 for ResNet18, in Tables 1, 2 and 3, respectively. Due to space limitation, the result on CIFAR100 for ResNet20 is left to Appendix A.3. As can be seen from the results in Tables 1 and 2, WaRP consistently outperforms all the baselines in whole incremental sessions. More specifically, iCaRL, Rebalance and EEIL are far below WaRP, since all these methods are not originally developed for handling the lack of data in CIFSL. The naive finetuning method shows poor performance in all results as the model is highly overfitted to new classes and forgets previous ones rapidly due to scarcity of data. It is worth noting that the performance gain of WaRP against one of SOTA methods, F2M, is 2.68% for *mini*ImageNet and 1.12% for CUB200. For CIFAR100 dataset, since there are no prior results for ResNet18, we reproduced the results of the following SOTA methods: GPM, FSLL, CEC and F2M. In Table 3, the trend is consistent with the results on *mini*ImageNet and CUB200, where the performance gains over the F2M and CEC are 0.75% and 1.30%, respectively.

To see the effect of finetuning in WaRP clearly, we compare our method with a simple and strong baseline which takes the prototype of each class as classifier for both base and novel classes. Since this baseline is equivalent to the case of our method with no finetuning, we can directly see the pure gains of finetuning in WaRP by comparing with this baseline. As can seen from the results on CUB200 in Table 4, this prototype-based baseline already outperforms or is comparable to SOTA methods, which is consistent with the observation from F2M (Shi et al., 2021). We can see performance gains

Table 4: Performance improvement of WaRP over prototype-based baseline on CUB200 dataset.

| Method | Session | | | | | | | | | | |
|---|---|---|---|---|---|---|---|---|---|---|---|
| | 1 | 2 | 3 | 4 | 5 | 6 | 7 | 8 | 9 | 10 | 11 |
| Baseline (Prototype) | 77.74 | 73.88 | 70.40 | 66.45 | 64.40 | 61.88 | 60.46 | 58.89 | 56.93 | 56.48 | 55.46 |
| WaRP (Ours) | 77.74 | 74.15 | 70.82 | 66.90 | 65.01 | 62.64 | 61.40 | 59.86 | 57.95 | 57.77 | 57.01 |
| Improvement | | +0.27 | +0.42 | +0.45 | +0.61 | +0.76 | +0.94 | +0.97 | +1.02 | +1.29 | +1.55 |

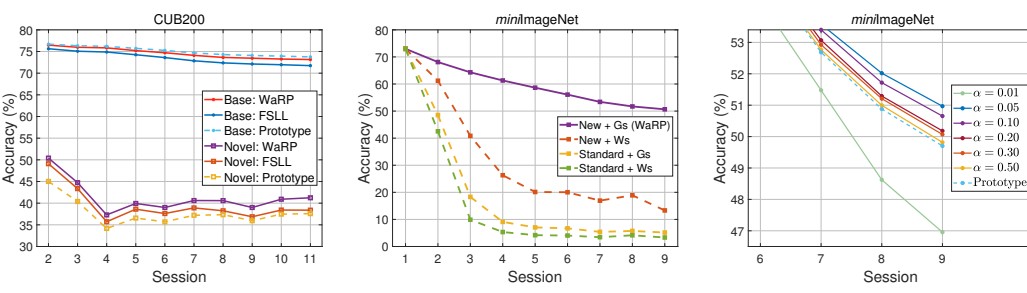

(a) Accuracy on base/novel classes.    (b) Effect of basis and score.    (c) Performance with varying $\alpha$.

Figure 3: Further studies on WaRP using CUB200 and *mini*ImageNet datasets.

of WaRP over this strong baseline coherently in all incremental sessions, where the gain increases as the session progresses.

## 5.3 FURTHER STUDIES ON WARP

**Performance on Base and Novel Classes.** In Figure 3a, we report the accuracies on base classes and novel classes separately. We compare our WaRP with FSLL (Mazumder et al., 2021), which also finetunes the model by selecting trainable parameters as ours, but in the original parameter space defined with the standard basis. We also report the performance of the prototype-based baseline considered in Table 4. We have the following key observations in Figure 3a. First, for all sessions, the base class performance of WaRP sacrifices little compared to the prototype baseline (0.61% drop in the last session) while achieving better result than FSLL (1.40% gain in the last session). This indicates that WaRP can effectively keep the knowledge of base classes throughout the learning process. Regarding the novel classes, WaRP has considerable gains compared to both FSLL (1.2%-2.8% gain) and the prototype-based baseline (3%-5% gain), indicating that WaRP effectively provides additional room for learning new classes in the new space, compared to FSLL which may limit the learnability on new classes due to additional regularization. The overall results confirm the ability of WaRP for preserving the knowledge of previous classes and learning new classes.

**Ablation on Choices of Basis/Score.** To see how the proposed new basis and score criterion work in a highly complementary manner, in Figure 3b, we compare the performance of 4 different (basis, score) combinations: we consider standard basis and new basis for the basis candidates. For the score criterion, we consider weight score (Ws) and gradient score (Gs), which are the magnitudes of weight and gradient, respectively. As we can see, if we do not apply our new space and score criterion simultaneously, the performance extremely degrades as the session progresses while our method shows much better performance than other combinations.

**Ablation on Varying $\alpha$.** In Figure 3c, we consider ablation on varying $\alpha$ ranging from 0.01 to 0.50 to see the robustness of WaRP against the perturbing hyperparameter. For an extreme case with $\alpha = 0.01$, the accuracy is low since most of model parameters are updated and thus the previous knowledge cannot be preserved. However, if $\alpha$ is larger than a specific small threshold (0.05 in this case), WaRP consistently provides additional gain against the baseline, suggesting that the performance gain is robust to perturbing $\alpha$.

## 6 CONCLUSION

In this paper, we introduced the concept of weight space rotation process (WaRP), a general framework for reparameterizing the weight matrix of the neural network. By taking advantage of WaRP, we proposed a novel strategy for CIFSL that effectively learns new classes while preserving previous knowledge, alleviating forgetting/overfitting issues. Experimental results on various benchmarks confirmed the advantage of our method over the state-of-the-arts. Although we focused on CIFSL in this paper, the general concept of WaRP can be utilized in any deep learning applications where the knowledge of the model should be kept in a few parameters, which we leave for a future work.

REPRODUCIBILITY STATEMENT

The detailed experimental setups including datasets, model architectures, evaluation metrics and hyperparameters are described in Sec. 5.1 and Appendix A.1. We also provide the source code in Supplementary Material. We run the simulations 5 times with different random seeds using NVIDIA GeForce RTX 3090 GPU machine and report average values. The code is available at https://github.com/EdwinKim3069/WaRP-CIFSL.

ACKNOWLEDGMENTS

This work was supported by Samsung Electronics Co., Ltd. and by IITP funds from MSIT of Korea (No. 2020-0-00626).

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

# A APPENDIX

## A.1 HYPERPARAMETERS

**Optimizer.** We use SGD optimizer with a momentum of $0.9$ throughout the whole sessions for all our simulations.

**First Session ($k = 1$).** We pretrain the model with the batch size of 128 in the first session for all datasets. The number of pretraining epochs is 210 for ResNet18 on both CIFAR100 and *mini*ImageNet and 300 for ResNet20 on CIFAR100. Regarding ResNet18 on CUB200, as we adopt Pytorch built-in pretrained model as done in most of previous works (Tao et al., 2020; Zhang et al., 2021), we pretrain the model using a smaller learning rate of 0.01 following (Chen & Lee, 2021), and the total number of epochs is 120. The learning rate decays by 1/10 at last 90, 60, 30 epoch to the end of training (for example, the learning rate decays at 210, 240, 270 if the total training epochs is 300).

**Incremental Sessions ($k \geq 2$).** Following (Mazumder et al., 2021), we finetune the model with training epochs of 30 for all datasets/architectures in WaRP experiments. We use the learning rate of 0.01 for ResNet18 and 0.001 for ResNet20. When searching the learning rate, we randomly split trainset of base classes into train/validation so that the validation set has a ratio of about 10% in total. We also randomly sample the data from trainset of novel classes (different from the 5-shots that we use for training) and use them as validation set for novel classes. We search the learning rate within the grids of [1e-1, 1e-2, 1e-3] on CUB200 for ResNet18 and on CIFAR100 for ResNet20. As for the *mini*ImageNet and CIFAR100 for ResNet18, we follow the setting of CUB200.

**Overall Settings for WaRP Algorithm.** Throughout the whole datasets/architectures, we randomly sample 20 batches ($20 \times 128 = 2,560$ samples) from base classes and use them for computing SVD in the first session. For computing score criterion, we set the batch size in equation (9) to be $n_b = 60$ for base classes and $n_b = 1$ for new classes. As for the layers to be applied weight space rotation, we apply WaRP to the last two resnet blocks (out of four blocks in total) for ResNet18 model and apply to the last one block (out of three blocks in total) for ResNet20 model. We do not apply WaRP to the batch-normalization layer. Note that we freeze all the layers, to which WaRP is not applied, after completing the first session. The hyperparameter of keeping ratio $\alpha$ at each session is set to be 0.1. The temperature for cosine similarity (Gidaris & Komodakis, 2018) in classifier is set to be a constant value of 16, following (Zhang et al., 2021).

## A.2 IMPLEMENTATION OF WaRP FOR CONVOLUTIONAL LAYER CASE

We first denote the filter (kernel) of convolutional layer by $W \in \mathbb{R}^{m \times n \times k \times k}$ where $k$ is size of kernel, $m$ is the target number of output channels and $n$ is the number of input channels, and let the activation (input of the layer) be denoted by $\Phi \in \mathbb{R}^{n \times h \times w}$ where $h$ and $w$ are height and width of activation feature map respectively. For computing SVD of activation map, all the patches in the feature map, that the kernel slides across, are flattened as a vector form and all these flattened vectors are stacked to be 2-dimensional matrix. For example, if the kernel size $k$ is 3 and the stride and padding are 1 respectively, the shape of stacked version of activation matrix is $(3 \times 3 \times n, h \times w)$ and the resulting unitary matrix would be $U_{\text{2-dim}} \in \mathbb{R}^{9n \times 9n}$. Then after we reshape the filter $W$ into 2-dimensional weight matrix as $W_{\text{2-dim}} \in \mathbb{R}^{m \times nk^2}$, the remaining process is the same with fully-connected layer case; we reparameterize the layer with $\widetilde{W}_{\text{2-dim}} = V_{\text{2-dim}}^\top W_{\text{2-dim}} U_{\text{2-dim}}$ where $V_{\text{2-dim}} \in \mathbb{R}^{m \times m}$ and $U_{\text{2-dim}} \in \mathbb{R}^{nk^2 \times nk^2}$ are unitary matrices.

During computing the loss, there are two options we can choose to implement $W \leftarrow V \widetilde{W} U^\top$. The first one is just to compute $W_{\text{2-dim}} = V_{\text{2-dim}} \widetilde{W}_{\text{2-dim}} U_{\text{2-dim}}^\top$ and reshape $W_{\text{2-dim}}$ to be the same with $W \in \mathbb{R}^{m \times n \times k \times k}$ at every loss computation. This approach is easy to implement but might be computationally inefficient. To pursue a slight computational efficiency, we take the second approach: configure this layer as three convolutions with filters of $V_{\text{conv}} \in \mathbb{R}^{m \times m \times 1 \times 1}$, $\widetilde{W}_{\text{conv}} \in \mathbb{R}^{m \times nk^2 \times 1 \times 1}$ and $U_{\text{conv}} \in \mathbb{R}^{nk^2 \times n \times k \times k}$ where $V_{\text{conv}}, \widetilde{W}_{\text{conv}}$ and $U_{\text{conv}}$ are reshaped version of $V_{\text{2-dim}}, \widetilde{W}_{\text{2-dim}}$ and $U_{\text{2-dim}}$ respectively.

## A.3 ADDITIONAL EXPERIMENTAL RESULTS

We report additional results on CIFAR100 for ResNet20 as shown in Table 5. As ResNet20 is too narrow (e.g., the number of channels in the last layer is 64) to make additional room that can be reaped by WaRP for learning new classes, the gain is rather minimal compared to the results for wider network of ResNet18 (the number of channels in the last layer is 512) as shown in Table 1, 2 and 3. However, as can be seen in Table 6, WaRP still consistently provides gains in all sessions against prototype-based baseline which we considered in Sec. 5.2. Note that the prototype-based baseline is strong in that this baseline already outperforms or is comparable to state-of-the-art methods, which is also confirmed in F2M (Shi et al., 2021). Moreover, the effectiveness of WaRP is not restricted by the dataset; result on the same dataset, CIFAR100, for ResNet18 achieves state-of-the-art as can be seen in Table 3 and reads appreciable gains compared to ResNet20 case as shown in Table 7. Thus, the limited gain in small network can be easily handled by replacing the layers of the model with wider layers in practical deployment of our method.

Table 8 and Figure 4 refers to the results on repeated experiments of Table 4 and Figure 3c on different datasets respectively. As shown in Figure 4, WaRP consistently provides the gains against strong baseline (prototype) once the $\alpha$ is larger than a certain threshold value, which suggests that the WaRP is fairly robust to perturbing hyperparameter $\alpha$ value.

Table 5: Accuracy on CIFAR100 dataset under 5-way 5-shot incremental few-shot learning setup for ResNet20.

| Method | Session | | | | | | | | |
|---|---|---|---|---|---|---|---|---|---|
| | 1 | 2 | 3 | 4 | 5 | 6 | 7 | 8 | 9 |
| Finetuning | 74.21 | 69.81 | 65.61 | 61.59 | 58.53 | 55.59 | 53.29 | 51.20 | 49.06 |
| iCaRL (Rebuffi et al., 2017) | 72.05 | 65.35 | 61.55 | 57.83 | 54.61 | 51.74 | 49.71 | 47.49 | 45.03 |
| Rebalance (Hou et al., 2019) | **74.45** | 67.74 | 62.72 | 57.14 | 52.78 | 48.62 | 45.56 | 42.43 | 39.22 |
| GPM (Saha et al., 2021) | 74.21 | 69.84 | 65.66 | 61.69 | 58.53 | 55.56 | 53.29 | 51.27 | 49.16 |
| EEIL (Castro et al., 2018) | 64.10 | 53.11 | 43.71 | 35.15 | 28.96 | 24.98 | 21.01 | 17.26 | 15.85 |
| TOPIC (Tao et al., 2020) | 64.10 | 55.88 | 47.07 | 45.16 | 40.11 | 36.38 | 33.96 | 31.55 | 29.37 |
| FSLL (Mazumder et al., 2021) | 64.10 | 55.85 | 51.71 | 48.59 | 45.34 | 43.25 | 41.52 | 39.81 | 38.16 |
| FSLL+SS (Mazumder et al., 2021) | 66.76 | 55.52 | 52.20 | 49.17 | 46.23 | 44.64 | 43.07 | 41.20 | 39.57 |
| CEC (Zhang et al., 2021) | 73.07 | 68.88 | 65.26 | 61.19 | 58.09 | 55.57 | 53.22 | 51.34 | 49.14 |
| F2M (Shi et al., 2021) | 71.45 | 68.10 | 64.43 | 60.80 | 57.76 | 55.26 | **53.53** | **51.57** | **49.35** |
| WaRP (Ours) | 74.21 | **69.96** | **65.86** | **61.92** | **58.74** | **55.79** | 53.50 | 51.51 | 49.33 |

Table 6: Performance improvement of WaRP over prototype-based baseline on CIFAR100 dataset for ResNet20.

| Method | Session | | | | | | | | |
|---|---|---|---|---|---|---|---|---|---|
| | 1 | 2 | 3 | 4 | 5 | 6 | 7 | 8 | 9 |
| Baseline (Prototype) | 74.21 | 69.83 | 65.67 | 61.72 | 58.55 | 55.62 | 53.30 | 51.27 | 49.11 |
| WaRP (Ours) | 74.21 | 69.96 | 65.86 | 61.92 | 58.74 | 55.79 | 53.50 | 51.51 | 49.33 |
| Improvement | | **+0.13** | **+0.19** | **+0.20** | **+0.19** | **+0.17** | **+0.20** | **+0.24** | **+0.22** |

Table 7: Performance improvement of WaRP over prototype-based baseline on CIFAR100 dataset for ResNet18.

| Method | Session | | | | | | | | |
|---|---|---|---|---|---|---|---|---|---|
| | 1 | 2 | 3 | 4 | 5 | 6 | 7 | 8 | 9 |
| Baseline (Prototype) | 80.31 | 75.54 | 71.22 | 67.02 | 63.73 | 60.65 | 58.30 | 56.11 | 53.84 |
| WaRP (Ours) | 80.31 | 75.86 | 71.87 | 67.58 | 64.39 | 61.34 | 59.15 | 57.10 | 54.74 |
| Improvement | | **+0.32** | **+0.65** | **+0.56** | **+0.66** | **+0.69** | **+0.85** | **+0.99** | **+0.90** |

Additionally, we further compare our method with more recent works (Zhou et al., 2022b; Chi et al., 2022; Hersche et al., 2022; Liu et al., 2022) on *mini*ImageNet dataset. Note that as (Hersche et al., 2022) adopted ResNet12 network for evaluating their method, we reproduced the results of (Hersche et al., 2022) by adopting ResNet18 to match the architecture. As shown in the table 9, we confirm our method outperforms all the baselines, except for (Zhou et al., 2022b), but still shows better performance in most sessions. Moreover, we found that (Akyürek et al., 2022) provided the results with ResNet12 architecture as done in (Hersche et al., 2022), thus we also evaluate our

Table 8: Performance improvement of WaRP over prototype-based baseline on *mini*ImageNet dataset.

| Method | Session | | | | | | | | |
|---|---|---|---|---|---|---|---|---|---|
| | 1 | 2 | 3 | 4 | 5 | 6 | 7 | 8 | 9 |
| Baseline (Prototype) | 72.99 | 68.07 | 64.09 | 60.83 | 58.06 | 55.38 | 52.68 | 50.88 | 49.70 |
| WaRP (Ours) | 72.99 | 68.10 | 64.31 | 61.30 | 58.64 | 56.08 | 53.40 | 51.72 | 50.65 |
| Improvement | | **+0.03** | **+0.22** | **+0.47** | **+0.58** | **+0.70** | **+0.72** | **+0.84** | **+0.95** |

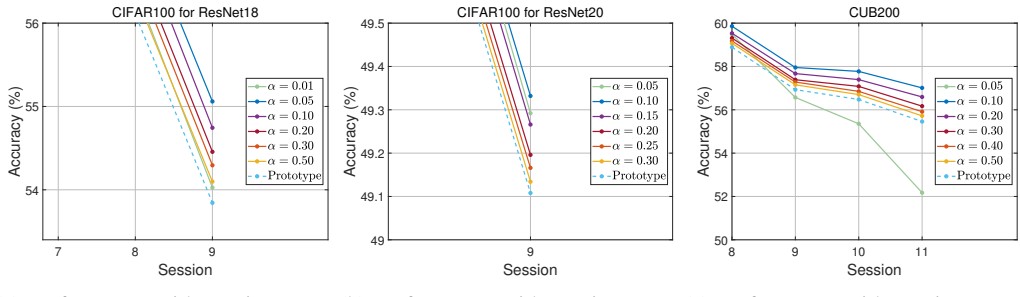

(a) Performance with varying $\alpha$ on CIFAR100 for ResNet18.

(b) Performance with varying $\alpha$ on CIFAR100 for ResNet20.

(c) Performance with varying $\alpha$ on CUB200.

Figure 4: Performance with varying $\alpha$ on different datasets/architectures.

method for ResNet12 architecture and compare with (Akyürek et al., 2022; Hersche et al., 2022) on *mini*ImageNet. As can be seen in the Table 10, our method shows better performance than the baselines which is consistent with the case for ResNet18.

Table 9: Accuracy on *mini*ImageNet dataset under 5-way 5-shot incremental few-shot setup for ResNet18.

| Method | Session | | | | | | | | |
|---|---|---|---|---|---|---|---|---|---|
| | 1 | 2 | 3 | 4 | 5 | 6 | 7 | 8 | 9 |
| LIMIT (Zhou et al., 2022b) | 72.32 | **68.47** | 64.30 | 60.78 | 57.95 | 55.07 | 52.70 | 50.72 | 49.19 |
| MetaFSCIL (Chi et al., 2022) | 72.04 | 67.94 | 63.77 | 60.29 | 57.58 | 55.16 | 52.90 | 50.79 | 49.19 |
| C-FSCIL Mode 1 (Hersche et al., 2022) | 64.72 | 59.63 | 55.43 | 51.95 | 48.99 | 46.44 | 43.86 | 41.76 | 39.78 |
| C-FSCIL Mode 2 (Hersche et al., 2022) | 64.73 | 59.68 | 55.41 | 52.04 | 49.31 | 46.65 | 44.11 | 42.09 | 40.15 |
| C-FSCIL Mode 3 (Hersche et al., 2022) | 64.73 | 59.75 | 55.23 | 51.97 | 49.05 | 46.36 | 43.62 | 41.43 | 39.69 |
| Entropy-reg (Liu et al., 2022) | 71.84 | 67.12 | 63.21 | 59.77 | 57.01 | 53.95 | 51.55 | 49.52 | 48.21 |
| WaRP (Ours) | **72.99** | 68.10 | **64.31** | **61.30** | **58.64** | **56.08** | **53.40** | **51.72** | **50.65** |

Lastly, we compare with (Hersche et al., 2022) on CIFAR100 dataset for ResNet12 (we further evaluate our method on CIFAR100 for ResNet12 instead of ResNet18), as (Hersche et al., 2022) also provided the results on CIFAR100 for ResNet12. Table 11 indicates that our method also shows the improved performance compared to (Hersche et al., 2022) on CIFAR100, which is consistent with the results on *mini*ImageNet.

## A.4 VALIDITY OF THE NEW BASIS ON NOVEL CLASSES

As aforementioned, once the new basis is constructed in the first session, the new basis is fixed during the whole remaining sessions. The future incoming tasks are not used to reconstruct or modify the basis to avoid unexpected behavior or statistical unreliability due to their lack of training samples. Here one may argue that the new basis is suitable only for preserving the knowledge of the base classes. Recall that we apply WaRP only to the last few layers while remaining the other layers fixed to resolve the overfitting issue. Here, since the activation of each layer depends not only on the task itself but also on preceding layers, the important parameters identified in the new basis are shared among the base and novel classes, i.e., the novel classes are also aligned with the new basis to some extent. To figure out this, we measure the keep ratio of important parameters that is accumulated as the session progressed. As shown in Figure 5, although we set keep ratio $\alpha$ as 0.1 for all sessions, the measured accumulated ratio does not grow by multiples of 0.1. Rather, the

Table 10: Accuracy on *mini*ImageNet dataset under 5-way 5-shot incremental few-shot setup for ResNet12.

| Method | Session | | | | | | | | |
|---|---|---|---|---|---|---|---|---|---|
| | 1 | 2 | 3 | 4 | 5 | 6 | 7 | 8 | 9 |
| Subspace-reg (Akyürek et al., 2022) | 80.37 | 73.92 | 69.00 | 65.10 | 61.73 | 58.12 | 54.98 | 52.21 | 49.65 |
| C-FSCIL Mode 1 (Hersche et al., 2022) | 76.37 | 70.94 | 66.36 | 62.64 | 59.31 | 56.02 | 53.14 | 51.04 | 48.87 |
| C-FSCIL Mode 2 (Hersche et al., 2022) | 76.45 | 71.23 | 66.71 | 63.01 | 60.09 | 56.73 | 53.94 | 52.01 | 50.08 |
| C-FSCIL Mode 3 (Hersche et al., 2022) | 76.40 | 71.14 | 66.46 | 63.29 | 60.42 | 57.46 | 54.78 | 53.11 | 51.41 |
| WaRP (Ours) | **82.05** | **77.06** | **73.17** | **70.19** | **67.92** | **65.14** | **62.36** | **60.57** | **59.49** |

Table 11: Accuracy on CIFAR100 dataset under 5-way 5-shot incremental few-shot setup for ResNet12.

| Method | Session | | | | | | | | |
|---|---|---|---|---|---|---|---|---|---|
| | 1 | 2 | 3 | 4 | 5 | 6 | 7 | 8 | 9 |
| C-FSCIL Mode 1 (Hersche et al., 2022) | 77.47 | 72.20 | 67.53 | 63.23 | 59.58 | 56.67 | 53.94 | 51.55 | 49.36 |
| C-FSCIL Mode 2 (Hersche et al., 2022) | 77.50 | 72.45 | 67.94 | 63.80 | 60.24 | 57.34 | 54.61 | 52.41 | 50.23 |
| C-FSCIL Mode 3 (Hersche et al., 2022) | 77.47 | 72.40 | 67.47 | 63.25 | 59.84 | 56.95 | 54.42 | 52.47 | 50.47 |
| WaRP (Ours) | **79.30** | **75.27** | **71.49** | **67.60** | **64.55** | **61.90** | **59.89** | **57.82** | **55.49** |

accumulated ratio increases by some value smaller than $\alpha$. As can be seen from the observation, the important parameters at each session somewhat overlap since the activation also depends on the preceding layers. This suggests that the new basis constructed in the first session would be aligned with the future tasks to a certain extent. Hence the new basis is also eligible for preserving the knowledge of the novel classes with our score criterion in (9).

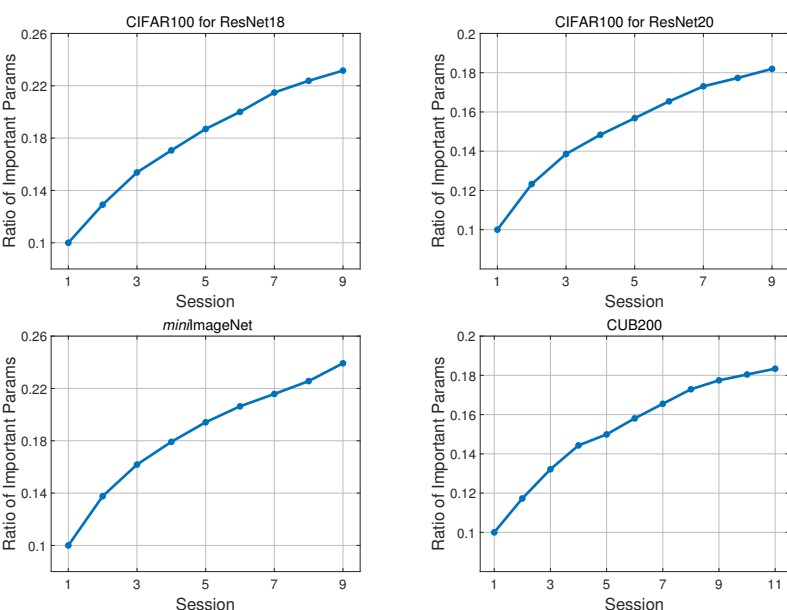

Figure 5: Accumulated keep ratios as the session progresses on various datasets.

Along with this observation, we further verify the ability to preserve the novel tasks by showing how the mode accuracy on novel classes varies as session grows. The table 12 and 13 show the accuracy on $\mathcal{T}_2$ and $\mathcal{T}_3$ respectively, i.e., the first and second incremental novel task. We evaluated on *mini*ImageNet dataset.

Moreover, to single out the effectiveness of WaRP in preserving the previous knowledge, we further evaluate the accuracies on a certain novel task when the model knows task ID. In other words, the prediction is made when the output logits corresponding to this novel task are only given. By doing so, we can exclude the effect of forgetting induced by the interference from the output logits

Table 12: Performance on the first novel task $\mathcal{T}_2$ at each incremental session on *mini*ImageNet dataset.

| Method | Session | | | | | | | |
|---|---|---|---|---|---|---|---|---|
| | 2 | 3 | 4 | 5 | 6 | 7 | 8 | 9 |
| Prototype | 21.52 | 20.68 | 19.88 | 19.28 | 18.92 | 18.24 | 17.84 | 17.04 |
| WaRP (Ours) | **27.16** | **25.48** | **24.32** | **23.92** | **23.52** | **22.84** | **22.12** | **21.04** |

Table 13: Performance on the first novel task $\mathcal{T}_3$ at each incremental session on *mini*ImageNet dataset.

| Method | Session | | | | | | |
|---|---|---|---|---|---|---|---|
| | 3 | 4 | 5 | 6 | 7 | 8 | 9 |
| Prototype | 22.32 | 21.48 | 21.00 | 19.68 | 18.48 | 17.96 | 17.88 |
| WaRP (Ours) | **27.76** | **26.76** | **25.84** | **23.64** | **22.00** | **21.60** | **21.52** |

corresponding to other tasks. As can be seen in the table 14 and 15, the new basis constructed using base classes is also suitable for preserving the knowledge of the novel classes.

Table 14: Performance on the first novel task $\mathcal{T}_2$ on *mini*ImageNet dataset when the task ID is known.

| Method | Session | | | | | | | |
|---|---|---|---|---|---|---|---|---|
| | 2 | 3 | 4 | 5 | 6 | 7 | 8 | 9 |
| Prototype | 58.92 | 58.92 | 58.92 | 58.92 | 58.92 | 58.92 | 58.92 | 58.92 |
| WaRP (Ours) | **60.60** | **60.76** | **60.68** | **60.56** | **60.36** | **60.40** | **60.40** | **60.48** |

## A.5 COMPARISON WITH EXISTING NETWORK PRUNING SCHEMES

As our method identifies the important parameters, we can compare our method against the network pruning schemes. We consider three network pruning methods: two for structured (channel/filter) pruning (Li et al., 2017; Liu et al., 2021) and one for unstructured pruning, SNIP (Lee et al., 2019). We adopted the methods of identifying parameters, including the score criterion and the structure of identification (e.g. whether identifying channel-wise or element-wise manner) proposed in these works. Although we set the ratio of important parameters $\alpha$ as 0.1 in WaRP experiments, during the reproducing the results on the methods from (Li et al., 2017; Liu et al., 2021; Lee et al., 2019), we tuned the parameter of this ratio $\alpha$ to get their best performances. We found that $\alpha = 0.9 \sim 0.95$ is suitable for (Li et al., 2017; Liu et al., 2021; Lee et al., 2019) as all of them still consider the original parameter space which exhibits the fundamental limitation as shown in the Figure 2. The results are evaluated on *mini*ImageNet dataset. As can be seen in the table 16, our method outperforms all of score criterion suggested in pruning schemes we considered, which indicates that the new space and its compatible score criterion are very effective in solving CIFSL.

## A.6 COMPUTATIONAL COST

We first separate the computational aspects of our scheme into two parts and discuss them separately: (a) constructing new basis in the first session, (b) finetuning/inference in the incremental sessions.

(a) After pretraining the model in the first session, we construct new basis by using singular value decomposition (SVD). The computational complexity of SVD is known as $O(n^3)$ where the covariance of activation is in the shape of $n \times n$. Although constructing the new basis appears to have somewhat large complexity due to the SVD, we only need to construct it *just once* in the first session. Moreover, the real time taken to construct the basis is not that significant compared to that of pretraining in the first session in practice. The table 17 shows the elapsed times for constructing the basis and pretraining with only 5 epochs on miniImageNet dataset. We tested on NVIDIA GeForce RTX 3090 GPU.

(b) During finetuning in the new basis, the complexity induced in the layer, to which the WaRP is applied, increases as much as when training with 3 layers due to $V$ and $U$ multiplied on both sides of

Table 15: Performance on the first novel task $\mathcal{T}_3$ on *mini*ImageNet dataset when the task ID is known.

| Method | Session | | | | | | |
|---|---|---|---|---|---|---|---|
| | 3 | 4 | 5 | 6 | 7 | 8 | 9 |
| Prototype | 69.40 | 69.40 | 69.40 | 69.40 | 69.40 | 69.40 | 69.40 |
| WaRP (Ours) | **71.04** | **70.72** | **70.80** | **70.64** | **70.64** | **70.52** | **70.44** |

Table 16: Comparison against various network pruning schemes on *mini*ImageNet dataset.

| Method | Session | | | | | | | | |
|---|---|---|---|---|---|---|---|---|---|
| | 1 | 2 | 3 | 4 | 5 | 6 | 7 | 8 | 9 |
| Filter prune (Li et al., 2017) (ratio=0.90) | 72.99 | 66.57 | 56.81 | 41.19 | 29.98 | 24.24 | 19.44 | 17.52 | 13.15 |
| Filter prune (Li et al., 2017) (ratio=0.95) | 72.99 | 67.15 | 62.07 | 55.77 | 49.75 | 44.88 | 38.39 | 36.17 | 33.14 |
| Channel prune (Liu et al., 2021) (ratio=0.90) | 72.99 | 66.77 | 61.19 | 56.78 | 53.05 | 49.34 | 45.39 | 42.75 | 40.37 |
| Channel prune (Liu et al., 2021) (ratio=0.95) | 72.99 | 67.81 | 63.75 | 60.58 | 57.86 | 55.26 | 52.50 | 50.71 | 49.55 |
| SNIP (Lee et al., 2019) (ratio=0.90) | 72.99 | 66.98 | 62.42 | 59.16 | 56.52 | 54.11 | 51.40 | 49.63 | 48.44 |
| SNIP (Lee et al., 2019) (ratio=0.95) | 72.99 | 67.65 | 62.41 | 58.38 | 55.48 | 53.04 | 50.38 | 48.59 | 47.48 |
| WaRP (Ours) | 72.99 | **68.10** | **64.31** | **61.30** | **58.64** | **56.08** | **53.40** | **51.72** | **50.65** |

$W$ in practical implementation. However, this difference does not that significantly slow down the finetuning as the model is finetuned only with a small iterations using only a few labeled samples in the actual incremental session. The table 18 shows the elapsed times for finetuning in both new and original space on miniImageNet dataset (we report averaged time per each incremental session). We tested on NVIDIA GeForce RTX 3090 GPU.

Moreover, we would like to highlight that, after completing the finetuning, we recover this layer to its original one, i.e., $W = V\widetilde{W}U^\top$ so that the inference time complexity does not increase.

### A.7 NOTES ON THE SPACE OF MATRIX

In equation (4), we deal with the "space of matrix" and generalize the linear algebraic concepts (which we are familiar with in space $\mathbb{R}^k$, for example) to this matrix space. Here, we define the inner product as $\langle A, B \rangle = \text{trace}(A^\top B)$ in Sec. 2. In fact, this inner product actually can be seen as a typical dot product in the vector space $\mathbb{R}^k$ for any $k$ (i.e. $\mathbf{a} \cdot \mathbf{b} = \sum_{i=1}^k a_i b_i$ for any vectors $\mathbf{a}, \mathbf{b} \in \mathbb{R}^k$ where $a_i$ and $b_i$ are $i$-th element of $\mathbf{a}$ and $\mathbf{b}$ respectively), if $A, B \in \mathbb{R}^{m \times n}$ are flattened as a vector form. In other words, it can be easily seen that $\langle A, B \rangle = A_{\text{flatten}} \cdot B_{\text{flatten}}$ where $A_{\text{flatten}}, B_{\text{flatten}} \in \mathbb{R}^{nm}$ is flattened version of $A$ and $B$ respectively. Thus, once $W$ and $\mathcal{K}_{ij}$ are flattened, we can easily interpret the equation (4) as the flattened version of $W$ and $\mathcal{K}_{ij}$ are in the vector space $\mathbb{R}^{nm}$ and the linear algebraic concepts we are familiar with in $\mathbb{R}^k$ such as linear combination, orthonormal vectors are directly adopted to this equation.

Table 17: Elapsed time for constructing new basis and pretraining the model with 5 epochs.

|                    | Construct basis | Pretrain with 5 epochs |
| ------------------ | --------------- | ---------------------- |
| Elapsed time (sec) | 136.73          | 123.49                 |

Table 18: Elapsed time for finetuning in new and original space.

|                    | New space | Original space |
| ------------------ | --------- | -------------- |
| Elapsed time (sec) | 1.2661    | 0.8472         |

