# OpenReview forum: "Warping the Space: Weight Space Rotation for Class-Incremental Few-Shot Learning"
_ICLR.cc/2023/Conference — ICLR 2023 notable top 25%_

### Official Review · Reviewer_hFPG · 2022-10-20

**Confidence:** 5
**Correctness:** 4
**Technical Novelty And Significance:** 2
**Empirical Novelty And Significance:** 2
**Recommendation:** 8

**Clarity, Quality, Novelty And Reproducibility:**

Clarity
=====
* For the most part the paper is clear. However, I found the description of the method complex and it looks like it could be simplified.

Quality
=====
* The quality of the related work could be improved. For example [B] presents a method to rotate the weights of a neural network to perform EWC, which is related to your work.

Novelty
======
* The proposed method is incrementally based on the work of Saha et al., 2021 and Wang et al., 2021, which the authors acknowledge (which is a positive thing).

Reproducibility
===========
* The authors provide all the necessary tools to reproduce their method.

[B] Liu, Xialei, et al. "Rotate your networks: Better weight consolidation and less catastrophic forgetting." 2018 24th International Conference on Pattern Recognition (ICPR). IEEE, 2018.

**Strength And Weaknesses:**

Strengths
=======
* The proposed method is sound and generalizes the methods of Saha et al., 2021 and Wang et al., 2021.
* Results are state of the art
* The authors provide ablations and code, which makes their work more reproducible

Weaknesses
=========
* Given the generality of the method, it is not clear why it is only tested on class-incremental learning
* Only papers from 2021 are considered for comparison, missing works like [A]
* I found the notation complex and I had to read the paper multiple times to understand all the equations. I still have trouble with the relationship between Hilbert spaces and equation 4.

[A] Hersche, Michael, et al. "Constrained Few-shot Class-incremental Learning." Proceedings of the IEEE/CVF Conference on Computer Vision and Pattern Recognition. 2022.




**Summary Of The Paper:**

The authors propose a method to rotate the weight space in a way that newly acquired knowledge does not interfere with the previous one. In the rotated space, only few dimensions are important for previous tasks and thus, they can be frozen to encode new knowledge in the remaining weights. The proposed method outperforms all the considered the state-of-the-art baselines.

**Summary Of The Review:**

The proposed method is sound and achieves good performance. My main concerns are about comparison with recent papers like [A], generality (why few-shot incremental and not class incremental, for instance), and the amount of novelty. Overall I think this is a good paper and I would be willing to raise my score after discussion.

---

> ### Author Response · Authors · 2022-11-18
> **Response to Reviewer hFPG (1/4)**
>
> We appreciate the reviewer for the valuable comments and constructive feedback. We  are  also thankful for the suggestions to add comparison with recent works and improvement of the related work section.
>
> &nbsp;
>
> ### **Missing comparison with recent works:**
>
> >We appreciate the reviewer's suggestion. In fact, the paper [A] that the reviewer has suggested, adopted a ResNet12 as the backbone network, which is different from the network (ResNet18) that the existing works (including our work) commonly adopted.  Hence for the fair comparison, we first adopt ResNet18 backbone and reproduce the results of [A] based on the official code. We evaluate [A] for ResNet18 on CIFAR100 and miniImageNet datasets, and compare with our method.
> >
> >&nbsp;
> >
> >- Accuracy on miniImageNet for ResNet18 at each session:
> >
> >| 　                   | Session 1 | Session 2 | Session 3 | Session 4 | Session 5 | Session 6 | Session 7 | Session 8 | Session 9 |
> |----------------------|-----------|-----------|-----------|-----------|-----------|-----------|-----------|-----------|-----------|
> | C-FSCIL Mode 1   [A] | 64.72%    | 59.63%    | 55.43%    | 51.95%    | 48.99%    | 46.44%    | 43.86%    | 41.76%    | 39.78%    |
> | C-FSCIL Mode 2   [A] | 64.73%    | 59.68%    | 55.41%    | 52.04%    | 49.31%    | 46.65%    | 44.11%    | 42.09%    | 40.15%    |
> | C-FSCIL Mode 3   [A] | 64.73%    | 59.75%    | 55.23%    | 51.97%    | 49.05%    | 46.36%    | 43.62%    | 41.43%    | 39.69%    |
> | WaRP (Ours)          | **72.99%**    | **68.10%**    | **64.31%**    | **61.30%**    | **58.64%**    | **56.08%**    | **53.40%**    | **51.72%**    | **50.65%**    |
> >
> >&nbsp;
> >
> >- Accuracy on CIFAR100 for ResNet18 at each session:
> >
> >| 　                   | Session 1 | Session 2 | Session 3 | Session 4 | Session 5 | Session 6 | Session 7 | Session 8 | Session 9 |
> |----------------------|-----------|-----------|-----------|-----------|-----------|-----------|-----------|-----------|-----------|
> | C-FSCIL Mode 1   [A] | 59.50%    | 55.31%    | 51.40%    | 48.11%    | 45.20%    | 42.39%    | 40.10%    | 37.88%    | 36.16%    |
> | C-FSCIL Mode 2   [A] | 59.48%    | 55.34%    | 51.60%    | 48.36%    | 45.49%    | 42.68%    | 40.40%    | 38.23%    | 36.46%    |
> | C-FSCIL Mode 3   [A] | 59.48%    | 55.17%    | 50.80%    | 47.43%    | 44.79%    | 42.12%    | 39.70%    | 37.68%    | 35.97%    |
> | WaRP (Ours)          | **80.31%**    | **75.86%**    | **71.87%**    | **67.58%**    | **64.39%**    | **61.34%**    | **59.15%**    | **57.10%**    | **54.74%**    |
> >
> >&nbsp;
> >
> >Although our method outperforms [A] on both datasets, we did not think it is still fair in that [A] starts from very low accuracy in the first session as shown in the table above. We did our best to match this accuracy, but did not make it. Thus, to pursuit a more fair comparison with [A], we conversely evaluate our method by adopting ResNet12 backbone. The following table shows the accuracies of both [A] and our method for ResNet12.
> >
> >&nbsp;
> >
> >- Accuracy on miniImageNet for ResNet12 at each session:
> >
> >| 　                   | Session 1 | Session 2 | Session 3 | Session 4 | Session 5 | Session 6 | Session 7 | Session 8 | Session 9 |
> |----------------------|-----------|-----------|-----------|-----------|-----------|-----------|-----------|-----------|-----------|
> | C-FSCIL Mode 1   [A] | 76.37%    | 70.94%    | 66.36%    | 62.64%    | 59.31%    | 56.02%    | 53.14%    | 51.04%    | 48.87%    |
> | C-FSCIL Mode 2   [A] | 76.45%    | 71.23%    | 66.71%    | 63.01%    | 60.09%    | 56.73%    | 53.94%    | 52.01%    | 50.08%    |
> | C-FSCIL Mode 3   [A] | 76.40%    | 71.14%    | 66.46%    | 63.29%    | 60.42%    | 57.46%    | 54.78%    | 53.11%    | 51.41%    |
> | WaRP (Ours)          | **82.05%**    | **77.06%**    | **73.17%**    | **70.19%**    | **67.92%**    | **65.14%**    | **62.36%**    | **60.57%**    | **59.49%**    |

---

> > ### Author Response · Authors · 2022-11-18
> > **Response to Reviewer hFPG (2/4)**
> >
> > &nbsp;
> >
> > ### **Missing comparison with recent works (continued):**
> >
> > >- Accuracy on CIFAR100 for ResNet12 at each session:
> > >
> > >| 　                   | Session 1 | Session 2 | Session 3 | Session 4 | Session 5 | Session 6 | Session 7 | Session 8 | Session 9 |
> > |----------------------|-----------|-----------|-----------|-----------|-----------|-----------|-----------|-----------|-----------|
> > | C-FSCIL Mode 1   [A] | 77.47%    | 72.20%    | 67.53%    | 63.23%    | 59.58%    | 56.67%    | 53.94%    | 51.55%    | 49.36%    |
> > | C-FSCIL Mode 2   [A] | 77.50%    | 72.45%    | 67.94%    | 63.80%    | 60.24%    | 57.34%    | 54.61%    | 52.41%    | 50.23%    |
> > | C-FSCIL Mode 3   [A] | 77.47%    | 72.40%    | 67.47%    | 63.25%    | 59.84%    | 56.95%    | 54.42%    | 52.47%    | 50.47%    |
> > | WaRP (Ours)          | **79.30%**    | **75.27%**    | **71.49%**    | **67.60%**    | **64.55%**    | **61.90%**    | **59.89%**    | **57.82%**    | **55.49%**    |
> > >
> > >As we can see in the tables for both ResNet12 and ResNet18, we confirm that our method consistently shows the better performances on both CIFAR100 and miniImageNet datasets. We have added these compared results to the Appendix A.3. Finally, as described in our response to Reviewer AjwX’s comment, we have also considered 5 additional baselines that were published in 2022. Please refer to our response above for more details.
> >
> > &nbsp;
> >
> > ### **Notations/descriptions of WaRP are complex:**
> >
> > >Thank you for pointing this out. To explain equation (4) simply, let us take any vector $p\in \mathbb{R}^4$ as an example. It is well known that, given the vectors $v_1, v_2, v_3, v_4 \in \mathbb{R}^4$ which are orthonormal, i.e. they are linearly independent and their norms are 1, then the vector $p$ can be always expressed as the linear combination of these vectors:
> > $$p=\alpha_1 v_1+\alpha_2 v_2+\alpha_3 v_3+\alpha_4 v_4$$
> > where $\alpha_1, \alpha_2, \alpha_3, \alpha_4$ can be easily obtained by the inner product of $p$ and the orthonormal vectors e.g., $\alpha_1 = p\cdot v_1$.
> > >However, "the space of the weight matrix" we are dealing with throughout the paper, is a bit different from the typical vector space we are familiar with. Thus, the reason we use the term 'Hilbert space', which is borrowed from analysis literature, is to generalize and explain the linear algebraic concepts (such as linear combination, orthonormal) in more general vector space equipped with properly defined inner product (e.g. vector space $\mathbb{R}^{m \times n}$ that the weight matrix $W$ lies in).
> > The inner product is up to our choice and we define it as $trace(A^{\top}B)$ in $\mathcal{H}=\mathbb{R}^{m \times n}$ throughout the paper. Here, one thing to note is that $trace(A^{\top}B)$ can be seen as a typical inner product (dot product) we are familiar with, if $A, B \in \mathbb{R}^{m \times n}$ are flattened as a vector form. Thus, thanks to our choice of the inner product, the example in $\mathbb{R}^4$ above can be extended/generalized to the equation (4) after flattening $W, \mathcal{K}$. We tried to make this point easier to understand and added the supplemental description in the revised manuscript. Due to the page limit during the discussion period, we put it in the Appendix A.7. But we will add this description in Section 2 after the page limit is lifted.
> >
> > &nbsp;
> >
> > ### **Improve the related work section:**
> >
> > >Thank you for your recommendation. We have added the paper that the reviewer has suggested with brief discussion to the related work section. This work also reparameterizes the weight as ours. However, we would like to emphasize that the role of 'rotate' is completely different from [B] in our context. The authors in [B] reparameterize the weight so that the Fisher Information Matrix (FIM) is as diagonal as possible. This reduces the information loss when the FIM is approximated by taking the diagonal only, when performing EWC. In our work, however, we rotate the axes of loss landscape so that we can find/utilize the flat directions, within which we can stably update the model without forgetting of the previous knowledge. We identify the important or flat directions that we freeze or update during finetuning by further exploiting the low-rankness property, which is also different from [B]. We tried to make these points clear in the revised manuscript. Please refer to our related work section for the details.

---

> > > ### Author Response · Authors · 2022-11-18
> > > **Response to Reviewer hFPG (3/4)**
> > >
> > > &nbsp;
> > >
> > > ### **Novelty beyond (Saha et al., 2021) and (Wang et al., 2021):**
> > >
> > > >We first want to highlight that our proposed method is not basically developed based on (Saha et al., 2021) or (Wang et al., 2021). Our intention of the expression “Note that our work generalizes the update process of (Saha et al., 2021; Wang et al., 2021)…” was not to say that we developed the algorithm based on (Saha et al., 2021) and (Wang et al., 2021), but to say that the update process of these works eventually turn out to be the special cases of our method.
> > > >Our work primarily starts from the general expression of the weight matrix in equation (4), to construct a ‘new space’ that tackles the fundamental limitation of the ‘original space’ shown in Fig. 2. Along with this, we also try to circumvent the limitations in the existing works such as F2M (Shi et al., 2021) and FSLL (Mazumder et al., 2021), where they find the 'flat region' in advance or identify/freeze the important parameters in the 'original space'. In our work, we construct a ‘new space’ (instead of 'original space') where we can find so many flat directions (instead of finding 'flat region') aligned with the new basis, and then make use of these ingredients (new space, flat directions) to effectively solve the CIFSL problem. We show that finetuning along these flat directions can effectively preserve the forgetting problem without dedicated process to find flat region, or imposing additional regularization on model parameters, which may limit the learnability of the model as shown in the Figure 3a in main manuscript.
> > > >
> > > >&nbsp;
> > > >
> > > >We also stress that the technical differences between ours and other CIL methods, including (Saha et al., 2021) & (Wang et al., 2021), make our scheme to be very effective in solving the overfitting issue, which is particularly significant in CIFSL problem.
> > > In our new weight space, the scores of identified unimportant parameters (which are much more than important parameters) are close to zero. Here, since perturbing these unimportant parameters has little effect on the model performance, it can be seen that only a few parameters in the new space are meaningful for determining the learnability of the model. In other words, the influences of most of parameters is compressed to only a few meaningful parameters in the new space. And recall that we apply WaRP only to the last few layers while remaining the other layers fixed. In this case, since the activation not only depends on the task itself but also depends on the preceding layers, the meaningful parameters are somewhat overlapped across tasks. Moreover, the deeper the layer, the more unimportant parameters are likely to be identified (close to zero). Thus, although we 'freeze only a few' parameters (*which is effective for preventing catastrophic forgetting*) during finetuning, we eventually 'update very small amount' of meaningful parameters (*which is effective for handling the overfitting issue*), which is not the case for both (Saha et al., 2021) and (Wang et al., 2021).
> > > As we can see in the table 1, 2, 3 where we have attached our reproduced values of (Saha et al., 2021) in the main manuscript, naively applying this scheme is not effective for tackling CIFSL, especially on CUB200 and CIFAR100 (underperform the simple prototype based baseline).
> > > >
> > > >&nbsp;
> > > >
> > > >Finally, we would like to summarize our novelty/contribution that makes our work unique from the existing works as follows:
> > > >- We introduced the concept of new basis of the weight space which enables us to readily identify the flat directions.
> > > >- We successfully identify such directions based on our proposed criterion, which is highly suited to our new weight space.
> > > >- We demonstrate there are many flat directions in the new space and finetuning along these flat directions can effectively preserve the forgetting issue, without finding the flat region in advance or applying the additional regularizations.
> > > >- We consolidate all the steps (constructing new basis, identifying flat directions, finetuning) so that our method solves the CIFSL problem well and achieves remarkable performance.

---

> > > > ### Author Response · Authors · 2022-11-18
> > > > **Response to Reviewer hFPG (4/4)**
> > > >
> > > >
> > > > &nbsp;
> > > >
> > > > ### **Despite the generality of WaRP, why it is applied only to CIFSL, not CIL. Moreover, is it really generalizable to other fields?:**
> > > >
> > > > >We first start by answering to why we specifically focused on CIFSL, instead of CIL. The CIFSL setup is similar to that of CIL with one notable difference that each incoming task contains only a few labeled samples, which makes the problem more practical yet extremely challenging to address. We focused on a challenging CIFSL problem as we found WaRP to be especially powerful/effective in handling both catastrophic forgetting and overfitting issues simultaneously (which is closely related to the answer in novelty part above). In other words, we thought that WaRP can maximize/show its advantage in CIFSL setup and provide a stronger impact into CIFSL community.
> > > > >
> > > > >&nbsp;
> > > > >
> > > > >Beyond CIL/CIFSL, the concept of WaRP can be applied to various other applications as we have argued in the discussion section. To support this claim, we tried to utilize WaRP to some important applications other than CIL/CIFSL. Although the slight algorithmic tuning is required to apply WaRP to the specific applications, we witnessed some potentials of WaRP in the fields of **Network Pruning** and **Gradient Compression**.
> > > > >
> > > > >&nbsp;
> > > > >
> > > > >In the **network pruning**, for example, we confirmed that removing most of unimportant parameters in the 'new space' (we conjectured that these unnecessary parameters in the new space bother pruning the network) improves the performance. By utilizing WaRP, we can keep only the meaningful parameters without degrading the accuracy (induced by parameter removing) as we push most of the knowledge compactly into only a few parameters. For some specific pruning algorithm (we apply magnitude based pruning in this case), applying the pruning scheme after keeping only meaningful parameters in the new space outperforms naively applying the pruning algorithm. The following table shows the performances when adopting WaRP before applying pruning method at a certain compression ratio (compression ratio := # of total elements / # of nonzero elements).  Mag refers magnitude based pruning, and Mag + WaRP means applying WaRP before Mag pruning. We finetune the model after pruning the network with the same epochs. The method is evaluated on CIFAR10 for ResNet34.
> > > > >&nbsp;
> > > > >
> > > > >| Compression   ratio | x128   | x256   | x512   | x1024  |
> > > > |---------------------|--------|--------|--------|--------|
> > > > | Mag                 | 82.45% | 71.48% | 61.86% | 47.30% |
> > > > | Mag  + WaRP         | **82.97%** | **76.04%** | **68.92%** | **61.24%** |
> > > > >
> > > > >&nbsp;
> > > > >
> > > > >Finally, we focused on **gradient compression**, which is commonly adopted to reduce communication burden in distributed learning where the gradient computed at each node is sent to the server. In fact, our WaRP rotate the axes so that most of directions aligned with new basis are flat and this means that only the remaining directions (parameters) are meaningful. Based on this insight, we can compress the gradient (in the new space side) by keeping the elements aligned with meaningful directions only, which is much less than the total number of elements. In other words, in the new space, the magnitudes of most elements of gradient w.r.t $\widetilde{W}$ are close to zero as can be seen in the equation (8) in the main manuscript. Thus, we can compress the gradient more aggressively by applying top-k to the gradient computed in the new space. The following table shows the number of communication bits required to reach the target accuracy. Our compression scheme is compared to naive top-k and no grad compression (denoted by 'Without comp' in the table). The server side constructs the new basis and broadcast to all the participating nodes (which we set as 100) to match the space on which the model is updated. The method is evaluated on CIFAR10 for VGG16 under non-iid setup and we set the target accuracy as 72%. We assume that the data is not shared among the clients due to the privacy concern, i.e., we specifically focused on federated learning setup.
> > > > >&nbsp;
> > > > >
> > > > >| 　                | Required number of bits |
> > > > |-------------------|:-----------:|
> > > > | Without comp      | 2.34E+11  |
> > > > | Naïve top-k       | 5.51E+09  |
> > > > | WaRP-based   comp | **2.53E+09**  |
> > > > >
> > > > >As we can see in the table, WaRP can improve the existing magnitude based pruning algorithm in network pruning and also enable us to aggressively compress the gradient in the new space. Further exploring the generality of the concept of WaRP in other applications is left to the future work.
> > > >
> > > > &nbsp;
> > > >
> > > > Again, we appreciate the reviewer for the time and efforts. Your concerns are clear and to the point, and we feel we have successfully clarified all the issues you raised. Specifically, we added more recent baselines, described the broad applicability of WaRP, and tried to make our novelty clearer. In case there are remaining questions/concerns, we hope to be able to have an opportunity to further answer them.

---

> ### Comment · Reviewer_hFPG · 2022-11-21
> **Response to authors**
>
> The authors made a clear and sincere effort to address all the reviewers' comments, resulting in significant improvement of their work. Thus, I am happy to raise my score to accept.

---

> > ### Author Response · Authors · 2022-11-22
> > **Thank you for the reply**
> >
> > Thank you very much for acknowledging our efforts and raising your score.

---

### Official Review · Reviewer_AjwX · 2022-10-24

**Confidence:** 5
**Clarity, Quality, Novelty And Reproducibility:** The paper is with a good clarity.
**Correctness:** 2
**Technical Novelty And Significance:** 2
**Empirical Novelty And Significance:** 2
**Recommendation:** 5

**Strength And Weaknesses:**

Strengths:

The paper is well-written. The motivation is clear and the method is technically correct.

Weaknesses:

1). In experiments, the comparisons with the previous methods are not enough. The following papers are not compared, but they seem to outperform the proposed method:

[1] Few-shot class incremental learning by sampling multi-phase tasks. (TPAMI)

[2] Subspace regularizers for few-shot class incremental learning. (ICLR ‘22)

[3] Metafscil: A meta-learning approach for few-shot class incremental learning. (CVPR ‘22)

[4] Constrained few-shot class-incremental learning. (CVPR ‘22)

[5] Few-shot class incremental learning via entropy-regularized data-free replay. (ECCV ‘22)

[6] Few-shot class-incremental learning from an open-set perspective. (ECCV ‘22)

I notice that the authors give references for [2] (introduction) and [1] (related work), but I do not find a reasons to ignore them in the comparison in experiments.

2). Fixing part of the network, though seems to be novel in FSCIL, but has been studied in the very similar task CIL (Adaptive Aggregation Networks for Class-Incremental Learning, CVPR ‘21). The motivations are very similar. It would be better to have a discussion instead of claiming “first work that makes changes to the parameter space itself”.


**Summary Of The Paper:**

The paper introduces weight space rotation process into the few-shot class-incremental learning. Specifically, the proposed method can identify the important parameters and freeze them in the following sessions.

**Summary Of The Review:**

The paper is well-written and has a clear motivation. But the experiments do not support the advantage of the proposed method. Comparison is not enough. Necessary discussion is lacking to support the novelty.

---

> ### Author Response · Authors · 2022-11-18
> **Response to Reviewer AjwX (1/2)**
>
> We appreciate the reviewer for the comments. We are also thankful for the criticisms including the suggestion for additional experiments.
>
> &nbsp;
>
> ### **Comparisons with other baselines:**
>
> >Thank you for pointing this out and providing suggestions for comparison with recent works. When comparing our method with other baselines, we tried to consider the methods having the same experimental setups with ours to seek to make our comparison as fair as possible. The experimental details, which the performance is highly dependent on, are slightly different across existing CIFSL algorithms. For example, one of the suggested works, [4] adopted a ResNet12 as the backbone network, which is different from the network (ResNet18) that the existing works (including our work) commonly adopted. Thus we adopt ResNet18 when reproducing the results of [4], based on the official code for fair comparison. As per the reviewer's suggestion, we tried to compare our method with 6 recent papers published in 2022 that the reviewer has suggested. Note that on CUB200 dataset, we further evaluate our method using another pretrained model that the authors of [1] provided and we denote it by WaRP*  in the table.
> >
> >&nbsp;
> >
> >- Accuracy on miniImageNet for ResNet18 at each session:
> >
> >| 　                   | Session 1 | Session 2 | Session 3 | Session 4 | Session 5 | Session 6 | Session 7 | Session 8 | Session 9 |
> |----------------------|-----------|-----------|-----------|-----------|-----------|-----------|-----------|-----------|-----------|
> | LIMIT [1]            | 72.32%    | **68.47%**    | 64.30%    | 60.78%    | 57.95%    | 55.07%    | 52.70%    | 50.72%    | 49.19%    |
> | MetaFSCIL [3]        | 72.04%    | 67.94%    | 63.77%    | 60.29%    | 57.58%    | 55.16%    | 52.90%    | 50.79%    | 49.19%    |
> | C-FSCIL Mode 1   [4] | 64.72%    | 59.63%    | 55.43%    | 51.95%    | 48.99%    | 46.44%    | 43.86%    | 41.76%    | 39.78%    |
> | C-FSCIL Mode 2   [4] | 64.73%    | 59.68%    | 55.41%    | 52.04%    | 49.31%    | 46.65%    | 44.11%    | 42.09%    | 40.15%    |
> | C-FSCIL Mode 3   [4] | 64.73%    | 59.75%    | 55.23%    | 51.97%    | 49.05%    | 46.36%    | 43.62%    | 41.43%    | 39.69%    |
> | Ent-reg [5]          | 71.84%    | 67.12%    | 63.21%    | 59.77%    | 57.01%    | 53.95%    | 51.55%    | 49.52%    | 48.21%    |
> | WaRP (Ours)          | **72.99%**    | 68.10%    | **64.31%**    | **61.30%**    | **58.64%**    | **56.08%**    | **53.40%**    | **51.72%**    | **50.65%**    |
> > &nbsp;
> >- Accuracy on CUB200 for ResNet18 at each session:
> >
> >| 　            | Session 1 | Session 2 | Session 3 | Session 4 | Session 5 | Session 6 | Session 7 | Session 8 | Session 9 | Session 10 | Session 11 |
> |---------------|-----------|-----------|-----------|-----------|-----------|-----------|-----------|-----------|-----------|------------|------------|
> | LIMIT [1]     | 75.89%    | 73.55%    | 71.99%    | 68.14%    | **67.42%**    | 63.61%    | **62.40%**    | **61.35%**    | **59.91%**    | 58.66%     | 57.41%     |
> | MetaFSCIL [3] | 75.90%    | 72.41%    | 68.78%    | 64.78%    | 62.96%    | 59.99%    | 58.30%    | 56.85%    | 54.78%    | 53.82%     | 52.64%     |
> | Ent-reg [5]   | 75.90%    | 72.14%    | 68.64%    | 63.76%    | 62.58%    | 59.11%    | 57.82%    | 55.89%    | 54.92%    | 53.58%     | 52.39%     |
> | WaRP (Ours)   | 77.74%    | 74.15%    | 70.82%    | 66.90%    | 65.01%    | 62.64%    | 61.40%    | 59.86%    | 57.95%    | 57.77%     | 57.01%     |
> | WaRP* (Ours)  | **78.88%**    | **75.79%**    | **72.68%**    | **68.70%**    | 66.93%    | **63.87%**    | 62.16%    | 61.31%    | 59.24%    | **58.81%**     | **57.72%**     |
> >
> >As shown in the table above, WaRP outperforms the recent baselines in all sessions, except for [1]. Compared with [1], our method still shows better performance in many sessions, including the last one (WaRP* in CUB200 table).

---

> > ### Author Response · Authors · 2022-11-18
> > **Response to Reviewer AjwX (2/2)**
> >
> >
> > ### **Comparisons with other baselines (continued):**
> >
> > >Moreover, we also evaluate our method using ResNet12 backbone network as we found that [2] provided the results with ResNet12 backbone in the paper as done in [4]. Thus we further compare our method with [2] and [4] on miniImageNet as shown in the table below.
> > >
> > >&nbsp;
> > >
> > >- Accuracy on miniImageNet for ResNet12 at each session:
> > >
> > >| 　                   | Session 1 | Session 2 | Session 3 | Session 4 | Session 5 | Session 6 | Session 7 | Session 8 | Session 9 |
> > |----------------------|-----------|-----------|-----------|-----------|-----------|-----------|-----------|-----------|-----------|
> > | Subs-reg [2]         | 80.37%    | 73.92%    | 69.00%       | 65.10%    | 61.73%    | 58.12%    | 54.98%    | 52.21%    | 49.65%    |
> > | C-FSCIL Mode 1   [4] | 76.37%    | 70.94%    | 66.36%    | 62.64%    | 59.31%    | 56.02%    | 53.14%    | 51.04%    | 48.87%    |
> > | C-FSCIL Mode 2   [4] | 76.45%    | 71.23%    | 66.71%    | 63.01%    | 60.09%    | 56.73%    | 53.94%    | 52.01%    | 50.08%    |
> > | C-FSCIL Mode 3   [4] | 76.40%    | 71.14%    | 66.46%    | 63.29%    | 60.42%    | 57.46%    | 54.78%    | 53.11%    | 51.41%    |
> > | WaRP (Ours)          | **82.05%**    | **77.06%**    | **73.17%**    | **70.19%**    | **67.92%**    | **65.14%**    | **62.36%**    | **60.57%**    | **59.49%**    |
> > >
> > >As can be seen in the table above, WaRP outperforms [2] and [4] on miniImageNet dataset for ResNet12 backbone which is consistent with the case for ResNet18.
> > >
> > >As for the comparison with [6], we also found that the backbone architecture adopted in [6] is slightly different from ours. So we evaluate our method using the same architecture copied from the official code that the authors of [6] provided for fair comparison. The following table shows the accuracy of both [6] and ours on miniImageNet when matching architecture.
> > >
> > >&nbsp;
> > >
> > >- Accuracy on miniImageNet for **ResNet18 copied from [6]** at each session:
> > >
> > >| 　          | Session 1 | Session 2 | Session 3 | Session 4 | Session 5 | Session 6 | Session 7 | Session 8 | Session 9 |
> > |-------------|-----------|-----------|-----------|-----------|-----------|-----------|-----------|-----------|-----------|
> > | ALICE [6]   | 80.60%      | 70.60%      | 67.40%      | 64.50%      | 62.50%      | 60.00%        | 57.80%      | 56.80%      | 55.70%      |
> > | WaRP (Ours) | **82.32%**    | **77.00%**    | **72.99%**    | **69.86%**    | **67.02%**    | **63.97%**    | **61.04%**    | **59.36%**    | **58.18%**    |
> > >
> > >Overall, we confirm that our method WaRP consistently outperforms or is comparable to other recent baselines when we match the experimental details such as backbone architecture.
> > In response to the reviewer's comment, we have added the comparisons with the papers published in 2022 to the revised manuscript. Please refer to the Appendix A.3 for more details.
> >
> > &nbsp;
> >
> > ### **Novelty compared to the existing work:**
> >
> > >As the reviewer said, the work (Adaptive Aggregation Networks for Class-Incremental Learning, CVPR 21) fixes part of the network during CIL. In fact, as described in the introduction and related works sections of our original manuscript, one of the existing works, FSLL (Mazumder et al., 2021) also fixes the part of the network (fixes some portions of parameters, specifically) during finetuning. However, our technical novelty does not lie in that point. We must stress that “fixing part of the network” and “changing the parameter space” are totally different concepts. These previous works (including the work that the reviewer has suggested) fix/update the model parameters, still in the "original parameter space”. The main message of our paper is that we can effectively facilitate CIFSL by fixing/updating the model in the “new parameter space”. More specifically, we introduced the concept of new basis of the weight space which enables us to readily identify the flat directions by exploiting the low-rankness property of activation. And then we successfully identify such directions based on our proposed criterion, which is highly compatible with our new weight space. We demonstrate there are many flat directions in the new space and finetuning along these flat directions can effectively preserve the forgetting issue, without finding the flat region in advance or applying the additional regularizations. We have tried to make these points clearer in the revised manuscript. Moreover, as per the reviewer’s comment, we have also tried to tone down our expression “first work that makes changes to the parameter space itself”. Please refer to our related work section for the details.
> >
> > &nbsp;
> >
> > Again, thank you for your time and efforts for reviewing our paper. Overall, we added more recent baselines, and added more discussions in the related work section. We hope that these are sufficient grounds for you to reconsider your rating. We would appreciate further opportunities to answer any remaining concerns you might have.

---

> > > ### Comment · Reviewer_AjwX · 2022-11-22
> > > **Thanks for the response**
> > >
> > > I thank the authors for providing the detailed response, especially for running the necessary experiments with fair backbone networks. The new experiment results could alleviate my concern about the performance comparison. But I think that these comparison results are necessary and should have appeared in the original submission. These results in the authors' response are significantly new contributions over the original submission. I am happy to raise my score though.

---

> > > > ### Author Response · Authors · 2022-11-22
> > > > **Thank you for the reply**
> > > >
> > > > We thank the reviewer for raising the score, but we would truly appreciate it if the reviewer could give yet another evaluation on the ground that 5 of the 6 new baselines we compared during rebuttal were published after May 28, 2022 and thus were not necessary for comparison, according to the official ICLR Q&A guideline. Note that we already had 8 baselines compared in our original submission. Please also consider the fact that our algorithm and message remained the same as we produced the additional experimental results that only strengthened the validity of our method.

---

### Official Review · Reviewer_1hev · 2022-10-25

**Confidence:** 3
**Correctness:** 3
**Technical Novelty And Significance:** 3
**Empirical Novelty And Significance:** 2
**Recommendation:** 6

**Clarity, Quality, Novelty And Reproducibility:**

**Clarity**

The narrative of the paper is clear and straightforward. Although the proposed method is somewhat complicated, authors do a great job in motivating each piece and explaining the underlying details of each component carefully. I appreciate the extra attention authors have paid to explain the overall procedure in a succinct manner (sec 3.3).

**Quality & Novelty**

I find this paper to be meeting the quality bar for a top-tier ML conference. The writing is crisp and pleasure to read, proposed method is well motivated and explained and finally, sufficient empirical results are provided to validate the hypothesis. The idea of moving the weight to a different basis and performing weight update in that space is unique and as a whole, there is enough novel contribution in this work.

**Reproducibility**

Authors have provided enough details pertaining to their proposed algorithm, backbone and other hyperparameters. My only minor concern regarding reproducibility is the difficulty in terms of extending this method to other neural network operators in future.




**Strength And Weaknesses:**

**Strengths**
* The narrative of the paper is crisp, easy to follow and enjoyable to read. Authors clearly motivate why changing the basis is important (instead of staying in the original parameter space) with an explanation in the two-dimensional space. After that, authors use the existing knowledge that activations reside in a low-dimensional sub-space to find unimportant parameters in this new basis and updates only those parameters for a given task while keeping the important parameters fixed. Every major component of the proposed algorithm is well motivated, clearly explained and intuitively makes sense.
* Apart from only providing a more theoretical overview of the proposed method, authors also provide details on how the weight-space change of basis (re-parameterization) can be implemented for a fully-connected and convolutional layer. Authors also improve the computational cost of the method by organizing the parameters in a way such that standard automatic differentiation packages can be used.
* Experimental results are performed on multiple FSCIL benchmarks and the proposed method outperforms existing methods (on-average, ~1 point over the SOTA method F2M). Authors also provide reasonable ablation studies to showcase the importance of various components including weight space re-parameterization, amount of forgetting with the amount of parameters modified etc. - overall, the empirical evaluations properly justify the efficacy of the algorithm.

**Weaknesses/Clarifications needed**
* For me, a major missing piece in this paper is the lack of comparison against methods from the domain of neural network pruning. Although the authors compare against existing methods from the literature, but for me, an important baseline to compare this method would be to take a pruning algorithm e.g. any channel pruning method for CNN, use that to identify which channels are unimportant after every round and only fine-tune those channels for a given task. Apart from channel pruning, other methods like layer pruning or structured pruning are also relevant to this discussion.
* Modifying the basis by re-parameterization requires non-trivial amount of work specific to each neural network operator. Authors provide guidance on how to do it for Conv/FFN, but I'd like to understand if this process can be generalized and applied to other kind of operators - multi-head attention from a Transformers network architecture or a transposed convolution operator for example. This can potentially hinder the adaptability of this method to other type of tasks or architectures.
* I'd like to see a plot on how many parameters are staying unimportant after every round - this will help us understand how does this method scale to more number of tasks.
* Finally, some details on the overall computational cost increase (at the beginning to change the basis and then for every iteration) compared to doing a standard fine-tuning will help us understand the cost/performance trade-off better.



**Summary Of The Paper:**

In this work, the authors propose an algorithm to efficiently learn new classes in an incremental setup with a few-shot dataset at every iteration. Towards this, they propose WaRP - which first changes the basis of the parameter/weight space of the base network and then uses SVD to find parameters with less importance in that space. At every round, a fraction of parameters which are found to be unimportant (with a metric defined in this paper) are fine-tuned to learn new classes while the important parameters are kept fixed. Important parameters found after each iteration are added to the original set of important parameters. On standard few-shot class incremental learning (FSCIL) benchmarks, WaRP outperforms existing methods.

**Summary Of The Review:**

Overall, I found the paper to be well written, clearly motivating the proposed method along with explaining each component in details and finally showcasing the efficacy of the method on standard benchmarks against current SOTA methods and via additional ablation studies. At this point, based on my opinion alone, I am leaning towards accepting this paper. I'd still like the authors to respond to my comments - especially regarding the channel pruning style baselines.

---

> ### Author Response · Authors · 2022-11-18
> **Response to Reviewer 1hev (1/2)**
>
> We  thank  the  reviewer  for  the  positive  comments  and  valuable  feedback.  We  appreciate  the reviewer’s acknowledgment on the uniqueness of our work. Our responses to the comments raised by the reviewer are given below.
>
> &nbsp;
>
> ### **Comparison against network pruning methods:**
>
> >Thank you for this suggestion. We compared our method against three network pruning schemes: two for structured (channel/filter) pruning [1, 2] and one for unstructured pruning, SNIP [3]. We specifically adopted the methods of identifying parameters, including the score criterion and the structure of identification (e.g. whether identifying channel-wise or element-wise manner) proposed in these works. Although we set the ratio of important parameters $\alpha$ as $0.1$ in WaRP experiments, when reproducing the results of the methods from [1, 2, 3], we tuned the parameter of this ratio $\alpha$ to get their best performances. We found that $\alpha=0.9$~$0.95$ is suitable for [1, 2, 3] as all of them still consider the original parameter space which exhibits the fundamental limitation as shown in Figure 2 in the main manuscript. The results are evaluated on miniImageNet dataset. As can be seen in the table below, our method outperforms all of score criterion suggested in pruning schemes we considered, which indicates that the new space and its compatible score criterion are very effective in solving CIFSL. We added this result in the Appendix A.5.
> >
> >| 　                   | Session 1 | Session 2 | Session 3 | Session 4 | Session 5 | Session 6 | Session 7 | Session 8 | Session 9 |
> |----------------------|-----------|-----------|-----------|-----------|-----------|-----------|-----------|-----------|-----------|
> | Filter Pruning [1]   (ratio=0.90)  | 72.99%    | 66.57%    | 56.81%    | 41.19%    | 29.98%    | 24.24%    | 19.44%    | 17.52%    | 13.15%    |
> | Filter Pruning [1]   (ratio=0.95)  | 72.99%    | 67.15%    | 62.07%    | 55.77%    | 49.75%    | 44.88%    | 38.39%    | 36.17%    | 33.14%    |
> | Channel Pruning [2]   (ratio=0.90)  | 72.99%    | 66.77%    | 61.19%    | 56.78%    | 53.05%    | 49.34%    | 45.39%    | 42.75%    | 40.37%    |
> | Channel Pruning [2]   (ratio=0.95)  | 72.99%    | 67.81%    | 63.75%    | 60.58%    | 57.86%    | 55.26%    | 52.50%    | 50.71%    | 49.55%    |
> | SNIP [3]   (ratio=0.90)  | 72.99%    | 66.98%    | 62.42%    | 59.16%    | 56.52%    | 54.11%    | 51.40%    | 49.63%    | 48.44%    |
> | SNIP [3]   (ratio=0.95)  | 72.99%    | 67.65%    | 62.41%    | 58.38%    | 55.48%    | 53.04%    | 50.38%    | 48.59%    | 47.48%    |
> | WaRP (Ours)          | 72.99%    | **68.10%**    | **64.31%**    | **61.30%**    | **58.64%**    | **56.08%**    | **53.40%**    | **51.72%**    | **50.65%**    |
> >
> >&nbsp;
> >
> >**References:**
> >
> >[1] Li, Hao, et al. "Pruning Filters for Efficient ConvNets", International Conference on Learning Representations, (ICLR) 2017.
> >
> >[2] Liu, Yuchen, et al. "Content-Aware GAN Compression",  Proceedings of the IEEE/CVF Conference on Computer Vision and Pattern Recognition, (CVPR) 2021.
> >
> >[3] Lee, Namhoon, et al. "Snip: single-Shot Network Pruning based on Connection sensitivity", International Conference on Learning Representations, (ICLR) 2019.
>
> &nbsp;
>
> ### **Adaptability of WaRP to other types of architectures:**
>
> >WaRP is applicable to any types of modules that consist of linear operations, including the multi-head attention module and the transpose convolutional layer. Whenever we can convert the underlying operation of the module to  matrix-matrix (vector) multiplication, the remaining process is exactly the same as we have done for fully-connected or convolutional layer. As for the multi-head attention, specifically, it generally comprises multiple linear transformations, each of which is typically implemented by the fully-connected layer, via 2-dimensional learnable parameters. Hence, WaRP can be applied to each head as done for fully-connected layer in this case. In addition, the underlying operation of the transpose convolutional layer is also linear. We can modify the input (activation) by properly introducing some intermediate zero pads to the input values so that the operation of transpose convolution is equivalent to that of the simple convolutional layer. In other words, applying WaRP to the transpose convolutional layer is exactly the same as done for simple convolutional layer except for proper modification to the activation.  Due to this broad applicability, we believe that WaRP can be adopted to various other applications that exploit these architectures (e.g. transformer in NLP tasks) and we hope that the strong adaptability of our scheme can be clearly conveyed through this discussion.

---

> > ### Author Response · Authors · 2022-11-18
> > **Response to Reviewer 1hev (2/2)**
> >
> > &nbsp;
> >
> > ### **Overall computational cost:**
> >
> > >We first separate the computational aspects of our scheme into two parts and discuss them separately: (a) constructing new basis in the first session, (b) finetuning/inference in the incremental sessions.
> > >
> > >(a) After pretraining the model in the first session, we construct new basis by using singular value decomposition (SVD). The computational complexity of SVD is known as $O(n^3)$ where the covariance of activation is in the shape of $n \times n$. Although constructing the new basis appears to have somewhat large complexity due to the SVD, we only need to construct it *just once* in the first session. Moreover, the real time taken to construct the basis is not that significant compared to that of pretraining in the first session in practice. The following table shows the elapsed times for constructing the basis and pretraining with only 5 epochs on miniImageNet dataset. We tested on NVIDIA GeForce RTX 3090 GPU.
> > >&nbsp;
> > >
> > >| 　                 | Constructing basis | Pretraining 5   epochs |
> > |--------------------|-----------------|--------------------------|
> > | Elapsed time | 136.73     sec     | 123.49    sec               |
> > >
> > >(b) During finetuning in the new basis, the complexity induced in the layer, to which the WaRP is applied, increases as much as when training with 3 layers due to $V$ and $U$ multiplied on both sides of $W$ in practical implementation. However, this difference does not that significantly slow down the finetuning as the model is finetuned only with a small iterations using only a few labeled samples in the actual incremental session. The following table shows the elapsed times for finetuning in both new and original space on miniImageNet dataset (we report averaged time per each incremental session). We tested on NVIDIA GeForce RTX 3090 GPU.
> > >&nbsp;
> > >
> > >| 　       | New space | Original space |
> > |----------|-----------|----------------|
> > | Elapsed time | 1.2661  sec   | 0.8472     sec    |
> > >
> > >Moreover, we would like to highlight that, after completing the finetuning, we recover this layer to its original one, i.e., $W = V\widetilde{W}U^{\top}$ so that the inference time complexity does not increase. We added these values in the Appendix A.6.
> > >
> >
> > &nbsp;
> >
> > ### **Accumulated important parameters & scalability to many tasks:**
> >
> > >Figure 5 in Appendix shows the ratio of accumulated important parameters (i.e., accumulated non-trainable parameters) in each incremental session. It can be seen that the important parameters at each session somewhat overlap with important parameters at other sessions, since the activation also depends on the preceding layers. In other words, the portion of trainable parameters decreases in a slow pace. Specifically, Figure 5 in Appendix A.4 refers to the portion of important parameters which are accumulated so far at each session. As can be seen in the Figure, the portion of accumulated important are less than 25% (i.e., portion of trainable parameters is more than 75%) for all three standard datasets (CIFAR100, miniImageNet, CUB200) after the whole multiple incremental sessions are completed. Below, we show Figure 5 in the tabular form (on miniImageNet as an example).
> > >&nbsp;
> > >| 　    | Session 1 | Session 2 | Session 3 | Session 4 | Session 5 | Session 6 | Session 7 | Session 8 | Session 9 |
> > |-------|-----------|-----------|-----------|-----------|-----------|-----------|-----------|-----------|-----------|
> > | Ratio | 0.1    | 0.1376    | 0.1618    | 0.1792    | 0.1941    | 0.2063    | 0.2158    | 0.2256    | 0.2392    |
> > >
> > >From our observation that a large number of trainable parameters remain even after many sessions, our method can be scalable to the case where a large number of tasks are sequentially provided.
> >
> > &nbsp;
> >
> > Overall, we agree with the reviewer’s comments and made efforts to discuss them in our response.  We hope to know if you are satisfied with our response. Please let us know if the reviewer has any further questions/concerns on our paper.

---

> > > ### Comment · Reviewer_1hev · 2022-11-18
> > > **Thanks for your rebuttal**
> > >
> > > Thanks for addressing my comments and appreciate all the efforts, especially in terms of running the pruning based baseline and compare your results against that. I'll be keeping my score and will be happy to defend this paper if situation demands.

---

> > > > ### Author Response · Authors · 2022-11-19
> > > > **Thank you for the reply**
> > > >
> > > > We appreciate your comment and particularly your willingness to defend our paper.

---

### Official Review · Reviewer_Z3CK · 2022-10-25

**Confidence:** 4
**Correctness:** 4
**Technical Novelty And Significance:** 3
**Empirical Novelty And Significance:** 3
**Recommendation:** 8

**Clarity, Quality, Novelty And Reproducibility:**

The clarity is excellent. In particular, the introduction and Figure 1 motivate the problem well. The quality is also good, with complete experiments and ablations. With respect to novelty, the paper does not have a large amount of strictly novel material per se but rather pieces together various bits of knowledge into an elegant solution for catastrophic forgetting. In my view, this is a sufficient amount of novelty. Reproducibility also is good and code is included in the supplementary materials.

Some other comments:
- For the ablations over alpha, I found it slightly disorienting for the plots to focus on the end sessions only. Perhaps showing the performance increase or decrease relative to a baseline alpha might make the plots easier to parse?
- One aspect of the submission that could be improved is an analysis of the learned basis and masks. For example, are parameters masked evenly throughout the network, or are they concentrated in either the lower or upper parts of the network? If the latter, then this matches the intuition that lower-level representations can be fixed with fine-tuning performed on top. WaRP offers data-driven masking and thus is in a unique position to be able to answer these questions.


**Strength And Weaknesses:**

Strengths
- Writing is clear and contains several interesting insights distributed throughout the paper.
- The method makes intuitive sense and should be easy to implement.
- The experiments are complete and contain a set of ablations justifying design decisions.
- The proposed WaRP method could have impact in a variety of transfer learning scenarios beyond CIFSL. Any setting where performance on previous tasks should be preserved can take advantage of the ideas behind WaRP.

Weaknesses
- The method seems to be primarily focused on preserving accuracy with respect to the base classes rather than the incrementally learned novel classes. In particular, the basis is computed only once after training on base classes, rather than updating after each session.
- After many learning sessions, it seems plausible that there could be a shortage of trainable parameters.

**Summary Of The Paper:**

This paper proposes a method to mitigate catastrophic forgetting in class-incremental few-shot learning (CIFSL). The key observation is that there may be many directions in the weight space that are flat with respect to the loss. If these directions can be identified, then updates may be freely made in these directions without affecting performance on the previous classes. In order to do this, a method called WaRP is proposed that utilizes the SVD of the network activations to construct a rotation such that the majority of directions are flat. For each subsequent incremental learning session, only a subset of the parameters with respect to the new basis are updated. The subset of parameters to be learned in each learning session are chosen based on a score criterion that relies on the magnitude of the gradients. Experiments on several standard CIFSL benchmarks demonstrate the improvements of the proposed WaRP method relative to baselines.

**Summary Of The Review:**

Update post-rebuttal: The authors have addressed my remaining concerns with additional results and ablations. I will maintain my original rating.

---

Overall, this is a strong submission. It addresses a meaningful problem and provides an effective and elegant solution to tackle it. Experiments show the proposed WaRP method provides consistent accuracy improvements relative to baselines.

---

> ### Author Response · Authors · 2022-11-18
> **Response to Reviewer Z3CK (1/3)**
>
> We appreciate the reviewer for acknowledging our work, and providing thoughtful comments. In particular, we are thankful for the recognition of our work being strong and novel. Below, we reply to the comments raised by the reviewer.
>
> &nbsp;
>
>
> ### **Basis is computed only once after training on the base classes:**
>
>
> > Interestingly, the basis constructed with the base classes also enables the model to preserve the knowledge of novel classes to some extent. Specifically, Figure 5 in Appendix shows the ratio of accumulated important parameters (i.e., accumulated non-trainable parameters) in each incremental session. It can be seen that the important parameters at each session somewhat overlap with important parameters at other sessions, since the activation also depends on the preceding layers. This suggests that the new basis constructed in the first session would be aligned with the future tasks to a certain extent. To verify this further, we show how the model accuracy on the novel classes varies as training session grows as shown in the tables below. We evaluate on miniImageNet dataset.
> >
> >- Accuracy on $\mathcal{T}_2$ (first incremental novel task):
> >
> >| 　          | Session 2 | Session 3 | Session 4 | Session 5 | Session 6 | Session 7 | Session 8 | Session 9 |
> |-------------|-----------|-----------|-----------|-----------|-----------|-----------|-----------|-----------|
> | Prototype   | 21.52%    | 20.68%    | 19.88%    | 19.28%    | 18.92%    | 18.24%    | 17.84%    | 17.04%    |
> | WaRP (Ours) | **27.16%**    | **25.48%**    | **24.32%**    | **23.92%**    | **23.52%**    | **22.84%**    | **22.12%**    | **21.04%**    |
> >
> >- Accuracy on $\mathcal{T}_3$ (second incremental novel task):
> >
> >| 　          | Session 3 | Session 4 | Session 5 | Session 6 | Session 7 | Session 8 | Session 9 |
> |-------------|-----------|-----------|-----------|-----------|-----------|-----------|-----------|
> | Prototype   | 22.32%    | 21.48%    | 21.00%    | 19.68%    | 18.48%    | 17.96%    | 17.88%    |
> | WaRP (Ours) | **27.76%**    | **26.76%**    | **25.84%**    | **23.64%**    | **22.00%**    | **21.60%**    | **21.52%**    |
> >
> >
> >Moreover, to single out the effectiveness of WaRP in preserving the previous knowledge, we further evaluate the accuracies on a certain novel task when the model knows task ID, as shown in the tables below. In other words, the prediction is made when the output logits corresponding to this novel task are only given. By doing so,  we  can  exclude  the  effect  of  forgetting  induced  by  the  interference  from  the  output  logits corresponding to other tasks.
> >
> >- Accuracy on $\mathcal{T}_2$ (first incremental novel task) when the task ID is known:
> >
> >| 　          | Session 2 | Session 3 | Session 4 | Session 5 | Session 6 | Session 7 | Session 8 | Session 9 |
> |-------------|-----------|-----------|-----------|-----------|-----------|-----------|-----------|-----------|
> | Prototype   | 58.92%    | 58.92%    | 58.92%    | 58.92%    | 58.92%    | 58.92%    | 58.92%    | 58.92%    |
> | WaRP (Ours) | **60.60%**    | **60.76%**    | **60.68%**    | **60.56%**    | **60.36%**    | **60.40%**    | **60.40%**    | **60.48%**    |
> >
> >- Accuracy on $\mathcal{T}_3$ (second incremental novel task) when the task ID is known:
> >
> >
> >| 　          | Session 3 | Session 4 | Session 5 | Session 6 | Session 7 | Session 8 | Session 9 |
> |-------------|-----------|-----------|-----------|-----------|-----------|-----------|-----------|
> | Prototype   | 69.40%    | 69.40%    | 69.40%    | 69.40%    | 69.40%    | 69.40%    | 69.40%    |
> | WaRP (Ours) | **71.04%**    | **70.72%**    | **70.80%**    | **70.64%**    | **70.64%**    | **70.52%**    | **70.44%**    |
> >
> >The tables shown above indicate that the basis constructed with the base classes is also eligible for preserving the knowledge of novel classes with our score criterion in (9). We have added this studies in Appendix A.4.

---

> > ### Author Response · Authors · 2022-11-18
> > **Response to Reviewer Z3CK (2/3)**
> >
> > &nbsp;
> >
> >
> > ### **It seems plausible that there could be a shortage of trainable parameters:**
> >
> > > This comment is also closely related to our answer above; since the important parameters somewhat overlap across different incremental sessions, the portion of trainable parameters decreases in a slow pace. More specifically, Figure 5 in Appendix A.4 shows the portion of important parameters which are accumulated so far at each session. As can be seen in the Figure, the portion of accumulated important are less than 25% (i.e., portion of trainable parameters is more than 75%) for all three standard datasets (CIFAR100, miniImageNet, CUB200) after the whole multiple incremental sessions are completed. Below, we show Figure 5 in the tabular form (on miniImageNet as an example).
> > | 　    | Session 1 | Session 2 | Session 3 | Session 4 | Session 5 | Session 6 | Session 7 | Session 8 | Session 9 |
> > |-------|-----------|-----------|-----------|-----------|-----------|-----------|-----------|-----------|-----------|
> > | Ratio | 0.1    | 0.1376    | 0.1618    | 0.1792    | 0.1941    | 0.2063    | 0.2158    | 0.2256    | 0.2392    |
> > >
> > > As we can see in the figure and table, there is still a room for ensuring a large number of trainable parameters, even after the multiple sessions.
> >
> > &nbsp;
> >
> > ### **Ablations over alpha:**
> >
> > > Below, we provide the performance with varying alpha for all sessions in the tabular form to see the effectiveness clearly on miniImageNet and CUB200 datasets. The results are consistent with the ones in our manuscript, indicating that the performance gain of the proposed idea is robust to perturbing alpha.
> > >&nbsp;
> > >
> > > - Performance with varying $\alpha$ on miniImageNet:
> > >
> > >| 　                     | Session 1 | Session 2 | Session 3 | Session 4 | Session 5 | Session 6 | Session 7 | Session 8 | Session9 |
> > |------------------------|-----------|-----------|-----------|-----------|-----------|-----------|-----------|-----------|----------|
> > | Ours   ($\alpha=0.05$) | 72.99%    | 68.10%    | 64.30%    | 61.39%    | 58.80%    | 56.28%    | 53.58%    | 52.02%    | 50.97%   |
> > | Ours   ($\alpha=0.10$) | 72.99%    | 68.10%    | 64.31%    | 61.30%    | 58.64%    | 56.08%    | 53.40%    | 51.72%    | 50.65%   |
> > | Ours   ($\alpha=0.20$) | 72.99%    | 68.12%    | 64.27%    | 61.02%    | 58.34%    | 55.66%    | 53.07%    | 51.29%    | 50.18%   |
> > | Ours   ($\alpha=0.30$) | 72.99%    | 68.12%    | 64.18%    | 60.97%    | 58.20%    | 55.56%    | 52.93%    | 51.21%    | 50.07%   |
> > | Ours   ($\alpha=0.50$) | 72.99%    | 68.05%    | 64.15%    | 60.90%    | 58.12%    | 55.44%    | 52.76%    | 51.00%    | 49.82%   |
> > | Prototype              | 72.99%    | 68.07%    | 64.09%    | 60.83%    | 58.06%    | 55.38%    | 52.68%    | 50.88%    | 49.70%   |
> > >
> > >
> > >- Performance with varying $\alpha$ on CUB200:
> > >
> > >
> > >| 　                     | Session 1 | Session 2 | Session 3 | Session 4 | Session 5 | Session 6 | Session 7 | Session 8 | Session 9 | Session 10 | Session 11 |
> > |------------------------|-----------|-----------|-----------|-----------|-----------|-----------|-----------|-----------|-----------|------------|------------|
> > | Ours   ($\alpha=0.10$) | 77.74%    | 74.15%    | 70.82%    | 66.90%    | 65.01%    | 62.64%    | 61.40%    | 59.86%    | 57.95%    | 57.77%     | 57.01%     |
> > | Ours   ($\alpha=0.20$) | 77.74%    | 74.09%    | 70.77%    | 66.81%    | 64.87%    | 62.39%    | 61.06%    | 59.53%    | 57.67%    | 57.39%     | 56.59%     |
> > | Ours   ($\alpha=0.30$) | 77.74%    | 74.05%    | 70.69%    | 66.69%    | 64.77%    | 62.21%    | 60.85%    | 59.31%    | 57.39%    | 57.08%     | 56.17%     |
> > | Ours   ($\alpha=0.40$) | 77.74%    | 73.96%    | 70.57%    | 66.70%    | 64.66%    | 62.13%    | 60.73%    | 59.18%    | 57.28%    | 56.85%     | 55.91%     |
> > | Ours   ($\alpha=0.50$) | 77.74%    | 73.93%    | 70.51%    | 66.58%    | 64.57%    | 62.02%    | 60.62%    | 59.07%    | 57.15%    | 56.70%     | 55.72%     |
> > | Prototype              | 77.74%    | 73.88%    | 70.40%    | 66.45%    | 64.40%    | 61.88%    | 60.46%    | 58.89%    | 56.93%    | 56.48%     | 55.46%     |

---

> > > ### Author Response · Authors · 2022-11-18
> > > **Response to Reviewer Z3CK (3/3)**
> > >
> > > &nbsp;
> > >
> > >
> > > ### **Additional studies on WaRP:**
> > >
> > > > Interestingly, when applying WaRP through the whole network, the early layers tend to have larger proportion of important parameter than the later ones.. To see this, we apply WaRP to the whole layers and  check the ratio of identified important parameters. We adopt miniImageNet dataset for ResNet18. ResNet18 model consists of 4 ResNet blocks and we show the ratios in block-wise manner. The tables below show that the ratio of identified important parameters  in ResNet block 3&4 is 0.0888 which is much less than that of the ResNet block 1&2 (ratio=0.2723). This indicates that the trainable parameters are somewhat concentrated on the later blocks. As per the reviewer’s comment, this result matches the intuition that the lower layers could be fixed with applying WaRP at the upper part of the network.
> > > >
> > > > | 　    | Block 1      | Block 2      | Block 3       | Block 4       |
> > > |-------|--------------|--------------|---------------|---------------|
> > > | Ratio |      0.2423  |      0.2807  |       0.2183  |       0.0565  |
> > > >
> > > > | 　    | Block 1&2    | Block 3&4    |
> > > |-------|--------------|--------------|
> > > | Ratio |      0.2723  |      0.0888  |
> > >
> > > &nbsp;
> > >
> > >  We again thank the reviewer for providing very helpful comments. We would love to answer any remaining concerns the reviewer might have.

---

> > > > ### Comment · Reviewer_Z3CK · 2022-12-08
> > > > **Thank you for your response**
> > > >
> > > > Thank you for your response and the updated results. In particular, the study on the ResNet blocks and the usage of trainable parameters over session are both quite interesting. I do not see any remaining issues and am happy to maintain my original rating.

---

### Decision · Program_Chairs · 2023-01-20

**Decision:**

Accept: notable-top-25%

**Justification For Why Not Higher Score:**

I think this paper, given its excellent clarity of presentation and crisp narrative, could make a decent oral presentation. In the end, however, I think it is somewhat lacking in terms of novel elements. I would not be disappointed to see it bumped up to an oral.

**Justification For Why Not Lower Score:**

The reviewers were nearly unanimous in their praise for the paper, for the clarity of presentation and the insights it provides through a sustained crisp narrative. The experimental results are equally well thought-out and excecuted.

**Metareview: Summary, Strengths And Weaknesses:**

# Summary of Contribution

This paper describes a method for Few-Shot Class-Incremental Learning that is based on identifying directions in parameter space that are "flat" with respect to the loss function. The authors propose to identify such directions and describe an approach (called WaRP) based on Singular Value Decomposition to represent the parameter space of the model in a basis in which most directions are "flat". Experiments on several standard few-shot class-incremental learning benchmarks demonstrate the improved performance of WaRP over baselines and state-of-the-art approaches.

# Strengths

+ **Clarity**: The paper is very well-written and provides clear motivations and insightful observations throughout all stages of the technical development. The narrative gracefully shifts between theoretical and technical levels of description with illustrations of how basis change in parameter space can be implemented for linear and convolutional layers. Reproducibility is high given the clear technical descriptions (and provided code).

+ **Experimental Results and Analysis**: The experimental results are thorough and the ablations provided illustrate the importance of each element of the contribution, including an analysis of forgetting as a function of the amount of parameters modified.

+ **Insights and Intuitions**: The paper contributed something interesting to the discussion on continual learning in general through its insights and methodical analysis of the effect of basis changes on few-shot incremental learning.

# Weaknesses

+ **Missing Comparisons with Recent Work**: Several reviewers commented that the paper lacks comparison with more recent work on few-shot continual learning. During the discussion phase the authors provided additional results (also justifying why it is difficult to compare with some published results due to architectural differences.

+ **Complexity**: Some reviewers commented at a few passages are difficult to comprehend on first reading, which however is perhaps to be expected given the theoretical nature of the central thesis of the contribution. In rebuttal the authors improved some aspects of this, and the supplementary appendices do a good job of providing additional details useful for deeper understanding.

# Summary

The reviewers are nearly unanimous in their positive opinion of this paper. The main outstanding concern was the lack of comparison with several more recent works on few-shot CIL. New results were provided during the discussion phase, which however leaves the question of whether or not these additional results constitute a significantly different work that that originally submitted. However, the positive aspects of this contribution greatly outweigh the negatives and it deserves a place at ICLR.


**Note From Pc:**

if the above contains the word "oral" or "spotlight" please see: "oral" presentation means -> notable-top-5% and "spotlight" means -> notable-top-25%. As stated in our emails, we are disassociating presentation type from AC recommendations